# A multi-dimensional analysis of native and non-native academic research articles in twelve disciplines

Jiaqi Deng[1,2], Ghayth Kamel Shaker Al-Shaibani[3]*

1 School of Foreign Languages, Southwest Medical University, Luzhou, Sichuan, China, 2 Department of Education, Faculty of Social Sciences and Liberal Arts, UCSI University, Kuala Lumpur, Malaysia, 3 Institute of Languages, UCSI University, Kuala Lumpur, Malaysia

* ghayth@ucsiuniversity.edu.my, ghayth.k@gmail.com

## Abstract

Multi-dimensional analysis (MDA) approach has been widely adopted to compare linguistic and register variations between native and non-native researchers' academic writing in numerous studies. However, only a few MDA studies have identified specific linguistic features for academic writing to develop an MDA framework suitable for this genre. To fill in this gap, this study identified 62 academic English linguistic features which were tagged with MDA tagger and counted by PatCount software programs to develop a novel MDA model for distinguishing native English and Chinese researchers' research article writing. This yielded four dimensions: academic involvement and interaction vs. information density; interactive argumentation vs. static description; impersonal evaluation vs. personal opinion; and explicit elaborating style vs. simplified reporting style. It was found that native English researchers are more academically and interactionally involved; they are more interactive and personal, and they use an explicit elaborating style than Chinese researchers do. In disciplines of hard or pure sciences, native researchers are more strategic to freely express their author stance, but Chinese researchers tend to be conservative as they follow pre-set disciplinary conventions. These findings suggest that Chinese researchers should exhibit their authorial stance, interact with the readers with confidence and employ more interactive devices to make their writing coherent and explicit. Meanwhile, Chinese conciseness displays clarity and efficiency for native researchers prone to verbosity; Chinese impersonality offers objectivity where natives risk bias. This study also proposed a new method to tag linguistic features in MDA research so that researchers can utilize this method to develop novel MDA models based on their research objectives.

**Data availability statement:** All relevant data are within the paper and its Supporting Information files.

**Funding:** Southwest Medical University, Luzhou, Sichuan, China Award Number: 2022YB024 | Recipient: Jiaqi Deng, MA The funder had no role in the study design, data collection and analysis, decision to publish, or preparation of the manuscript.

**Competing interests:** The authors have declared that no competing interests exist.

## 1. Introduction

Over the past 10 years, a considerable amount of research [1–6] on disciplinary variation between native English researchers and non-native researchers has been conducted. Understanding these differences helps non-native researchers meet the standards of English-speaking academic writing communities and enhances their writing skills to publish in reputable international journals [7]. Biber's (1988) [8] multi-dimensional analysis (MDA) model, which is favored by many researchers, stands out for its indispensable role in analyzing linguistic variations since it involves both macroscopic and microscopic perspectives. The macroscopic perspective refers to an MDA approach that identifies textual dimensions after analyzing the co-occurring patterns of linguistic features in selected research articles, whereas the microscopic perspective refers to interpreting these dimensions in functional terms by observing a large quantity of linguistic features in a substantial number of texts simultaneously.

The MDA model is based on the assumption that strong co-occurrence patterns of linguistic features mark underlying functional dimensions. Linguistic features do not randomly co-occur in texts. If certain features consistently co-occur, it is reasonable to look for an underlying functional influence that encourages their use [8]. Our MDA model has been constructed through identifying, tagging, counting linguistic features, then conducting a factor analysis which was widely used in many studies to distinguish language variation. Additionally, when comparing the differences in academic writing, only few MDA studies have identified linguistic features in academic writing by only adopting Biber's (1988) tagger [8] which was originally used to distinguish spoken and written linguistic features. Therefore, this study extends the previous studies in identifying linguistic features in academic writing, tagging and extracting these features through code writing and PatCount software, and eventually developing a novel MDA model. Furthermore, few previous studies have compared the linguistic variation in different disciplines between native English researchers and Chinese researchers via MDA model. In this study, a total of 2400 English research articles from 12 disciplines written by native English and Chinese researchers were collected to answer the following three research questions:

(i) What are the dimensions that may identify the lexical and grammatical differences between native English and Chinese researchers' academic writing in 12 disciplines?

(ii) What are the differences in the academic writing of native English and Chinese researchers across the 12 disciplines along these dimensions?

(iii) To what extent do language background (native English researchers vs Chinese researchers) and discipline influence the dimension scores of native English and Chinese researchers' academic writing?

## 2. Literature review

### 2.1 Discipline-based studies on research articles

Research articles are very common for scholars to communicate their findings to the academic community, adhering to specific disciplinary conventions and expectations

of the target academic discipline in which they are written [9]. Writing differs across disciplines meanwhile different academic disciplines shape writing practices [10,11]. Exploring how academic writing varies across disciplines helps non-native writers know how to adapt to these conventions [12]. While reviewing the previous research, two main trends were found. One was to compare the differences in linguistic features such as lexical choices, syntactic complexity, hedging and stance, audience awareness and rhetorical structures among different academic disciplines [13,14], and the other is conducted in subfields within one discipline. For instance, how various subfields use language differently within the field of applied linguistics [15], how sociolinguistics and language documentation differ in approaching linguistic variation within linguistics [16]; and metadiscourse choice variation within the field of English for academic purposes [17].

Three main perspectives were considered in these investigations: the macro perspective, the micro perspective, and the multi-dimensional perspective. A comparison of linguistic variation between native and non-native speakers from a macro perspective has usually been conducted via functional analysis of various genres, namely, move analysis [18]; a micro perspective has focused mostly on single lexical features, such as critical stances and evaluations [19], abstracts [20], the use of first personal pronouns [21], and hedges [22]. In addition, the MDA model has attracted the researchers' attention because it combines the advantages of the above-mentioned perspectives and has been applied to compare linguistic variation both macroscopically and microscopically. Studies on MDA are reviewed in detail in the following section.

## 2.2 MDA development and application

The origins of the MDA research can be dated back to the 1970s when the importance of the co-occurrence of linguistic features in comparative studies of registers attracted researchers' attention. Carroll (1960) provided a methodological basis for MDA by employing a statistical analysis of linguistic co-occurring patterns and performing all linguistic analyses manually to conduct a study on vectors of prose style. After that, Douglas Biber improved MDA in 1984 in his PhD dissertation. Biber selected 23 categories of 481 texts covering the spoken and written registers from the Lancaster-Oslo-Bergen (LOB) Corpus of British English and the London-Lund Corpus (LLC) of Spoken English in the UK, and he identified 67 linguistic features. Through statistical methods as factor analysis, Biber performed a multi-dimensional comparative analysis of spoken and written English of the co-occurrence of these linguistic features to identify seven dimensions [8]. The MDA represents a groundbreaking method for linguistic variation. It allows researchers to handle large corpora to identify patterns and dimensions of variation across different texts and registers by analyzing the co-occurring patterns of linguistic features via computational techniques. This approach is instrumental in understanding how language varies systematically according to various situational contexts.

The early MDA model was used to investigate the variation between oral and written English [8], and was widely used for other different registers [23,24]. After that, studies applied this approach to examine register variation in other languages, such as in Somali [25]. MDA subsequently experienced diachronic language variation research [26,27], and dialect variation research [28,29].

Two main paradigms have been employed in many studies comparing linguistic variations using MDA. One used Biber's (1988) [8] framework [30,31]. The other one developed a novel MDA suitable for one's research domain [32–34].

For these two paradigms, the one adopting Biber's (1988) [8] framework conducted the study under Biber's original framework with those linguistic features distinguishing spoken and written registers even in specialized disciplines. This approach would be less effective in analyzing variation in more restricted discourse domains. Thus, the other paradigm was commonly agreed to be more suitable especially for studies on a specialized discipline. However, using this paradigm requires selecting linguistic features for a specific discipline, tagging those features, counting the frequencies, and performing new factor analysis to identify new dimensions. It is tough for Applied Linguistics researchers with limited computer skills to tag those linguistic features related to their research, thus most studies use Biber's (1988) tag [8] and Biber's (2006) tag [35]. Biber's (1988) tag [8] is not freely accessible online; however, Biber's (1988) tag [8] was completely replicated by Nini [36] and can be accessed through multi-dimensional analysis tagger (MAT) programming for

free. Biber's (2006) tag [35] added additional semantic categories to Biber's (1988) tag [8] which is not freely available either, but may be accessed by contacting Douglas Biber's Corpus Linguistics Research Program at Northern Arizona University [37]. This study solved the difficulty of tagging specific linguistic features by using Nini's MAT and PatCount computer programming through the writing of computer codes and developed a novel MDA model related to academic writing in articles.

## 3. Methods

This section presents our research design in terms of data collection and data analysis in the sub-sections below resulting in developing a novel MDA model.

### 3.1 Data collection

This research adopts Biber's (1988) MDA development procedure: (1) the use of computer-based text corpora; (2) the use of computer programs to count the frequency of certain linguistic features in a large number of texts, enabling analysis of the distribution of many linguistic features across many texts; (3) the use of factor analysis to determine co-occurrence relationships among the linguistic features; (4) the use of microscopic analysis to interpret the functional parameters underlying the quantitatively identified co-occurrence patterns; and (5) a comparison of texts with respect to dimensions on the basis of computed factor scores [8].

To collect the research articles written by native English researchers and Chinese researchers in various disciplines, the disciplines to be studied in this research should be determined first. The criteria for selecting the disciplines followed the *Chinese National System of Level One Disciplines for Degree Education*, namely, 13 disciplines as agriculture, art, economics, history, law, literature, management science, medicine, natural science, education, philosophy, engineering and military. To make the native English researchers' corpus and Chinese researchers' corpus comparable, we selected the same disciplines and the same number of research articles for both corpora. When we checked native English researchers' disciplines in the Web of Science (WoS), there was no *military* discipline. Therefore, the number of final disciplines involved in this study was 12 when the *military* discipline was excluded. After the 12 disciplines were selected, we adopted stratified random sampling to select the research articles (RAs) in these 12 disciplines [38]. First, we used Journal Citation Reports (JCR) of the Web of Science (WoS) to look for the most influential ten international Science Citation Index (SCI), Social Science Citation Index (SSCI) and Science Citation Index Expanded (SCIE) journals in each discipline to select RAs based on the highest impact factor. Five of those ten journals in each discipline were then randomly selected. To select native English researchers' RAs, we chose the top international journals with the highest impact factor because they could be set as a benchmark. Twenty full-text articles from each journal for the past five years (2019–2023) were selected and included as data. The sample size for native English researchers' RAs is 1200. For native researchers' identification, it was not feasible for a study of this scale to directly contact all authors to confirm their L1 identity, thus we followed a widely adopted method in Applied Linguistics and English for Academic Purposes (EAP) research using a multi-tiered screening method of first author's affiliated institution identification and name identification [33,39–42]. First, the nation of the institution affiliated with which the paper is published was determined. Only institutional affiliation belongs to nations in the inner circle of English-speaking countries [43], specifically the United States, the United Kingdom, Canada, Australia, and New Zealand were included. Articles with which the nationality of the institution cannot be determined or do not belong to these five English-speaking nations were discarded. Then we determined whether the first author's nationality was matched with that of their publishing institution, by using a name origin database [44]. If the ethnic origin of the first author's surname was consistent with the nationality of the publishing institution, the author was regarded as originating from that nation. Articles that did not meet this criterion were excluded from the final corpus.

For the collection of Chinese researchers' RAs, the RAs were all chosen randomly from academic English journals published in mainland China whose authors were Chinese and were affiliated with an institution in mainland China to reflect

 

the Chinese researchers' average English writing level. All the English academic journals published in China were chosen according to Bao and Zhang [45]. The number of journals and articles was the same as that in the native researchers' corpus to ensure that the two corpora are balanced. The sample size was 1200 as well. These two corpora were named as the Native Researchers Corpus (NRC) and the Chinese Researchers Corpus (CRC). Both corpora comprised the same academic disciplines to avoid potential disciplinary biases in the selection process. Both corpora were collected from June 2024 to August 2024 and the total sample size of the research articles collected in this research is 2400. This sample size is sufficient for performing this research because in a factor analysis, the database should include five times as many texts as linguistic features to be analyzed [46]. In our study, 62 linguistic features were ultimately identified and included; thus, we have a sufficient 2400 articles as a sample for this study. The composition of the corpora including the total number of words for each discipline is shown in Table 1.

Most of the collected RAs are in PDF format. ABBYY FineReader, Adobe Acrobat Pro and other software were used to convert PDF format files into TXT format files. Two rounds of manual verification were carried out on the converted text to clean up the problems such as garbled code, formatting and spelling errors in the conversion process. Moreover, all the examples, directly quoted paragraphs, interview excerpts in the RAs and other languages that were not written by the author(s) were deleted. The quoted languages in the RAs do not represent the real language level of Chinese and English native researchers have been removed because retaining them may affect the accuracy of the results. Data such as charts and chart captions, which cannot reflect linguistic characteristics in the text, were deleted as well.

## 3.2  Data analysis

Textual analysis is a method of transforming textual data into quantifiable forms, revealing patterns and trends through classification, coding, and statistical analysis of texts [47]. Therefore, this study used textual analysis to identify and tag the linguistic features suitable for academic writing in the collected research articles, quantify the frequency of co-occurring linguistic features, and interpret these co-occurring dimensions after conducting a factor analysis.

**3.2.1  Identifying, tagging and counting the linguistic features.**  We determined that all linguistic features from Biber's (1988) tag [8] should be included as part of the linguistic features for this study because Biber's (1988) tag [8] is for spoken and written linguistic features and some Chinese researchers' academic writing is marked with spoken

**Table 1.  Composition of corpora.**

| Discipline | English native researchers | | Chinese researchers | |
|---|---|---|---|---|
| | No. of articles | Tokens | No. of articles | Tokens |
| Agriculture | 100 | 824200 | 100 | 557500 |
| Art | 100 | 516800 | 100 | 733600 |
| Economics | 100 | 1077500 | 100 | 675000 |
| Engineering | 100 | 729600 | 100 | 236400 |
| History | 100 | 1331400 | 100 | 793300 |
| Law | 100 | 718900 | 100 | 933800 |
| Literature | 100 | 800700 | 100 | 1010100 |
| Management science | 100 | 1015100 | 100 | 693000 |
| Medicine | 100 | 297200 | 100 | 362000 |
| Natural science | 100 | 423000 | 100 | 287200 |
| Education | 100 | 1109700 | 100 | 776200 |
| Philosophy | 100 | 278700 | 100 | 745100 |
| Total | 1200 | 9122800 | 1200 | 7803200 |

language (i.e., informal). Additionally, we included academic linguistic features for this study after reviewing the literature on academic writing research. We selected attitude markers [9,48], identity markers [9,49], boosters [9,50], reporting verbs [51,52], noun-noun phrases [53,54] and transition signals [55] which were the most investigated academic linguistic features in previous research. All the above-mentioned linguistic features for academic writing together with Biber's (1988) tag [8] constituted 73 new linguistic features considered for this study (see S5 Appendix).

After the linguistic features were identified, tagging these linguistic features was a vital and challenging step in developing a novel MDA. The linguistic features adopted from Biber [8] could be tagged through Biber's (1988) tag [8], but Biber's (1988) tag [8] is not accessible online. However, Nini developed a multi-dimensional analysis tagger (MAT) [36] by accurately replicating Biber's (1988) tag [8], and this program can be accessed online and downloaded. Therefore, for the linguistic features involving Biber's (1988) tag [8], we used Nini's MAT [36] for tagging. For those selected academic linguistic features, we first used the Stanford Part of Speech (PoS) tagger to tag all the words in the RAs, and then used PatCount software to extract and count the linguistic features selected in this study. The Standford PoS tagger is a software program that assigns parts of speech, such as nouns, verbs, and adjectives, to each word (and other token) of each text. When each word of every text was tagged by PoS, any linguistic feature can be extracted via the pattern code through PatCount. PatCount is a software program developed by the Language Engineering Laboratory of China Foreign Language Education Research Center at Beijing Foreign Studies University [56] and can be downloaded online and used for free of charge. It functions by using pattern matching technology in natural language processing to automatically analyze a corpus. By using pattern matching software through regular expressions, it is convenient to count the frequency of multiple linguistic features in a corpus. In this study, the codes of regular expressions of the selected academic linguistic features were written and input into PatCount, and then the frequency of each linguistic feature was counted through the written pattern files. For instance, *absolute*, a linguistic feature of boosters, was tagged by the Stanford PoS tagger as *absolute_JJ* and its regular expression was written as \sabsolute_JJ. When the code \sabsolute_JJ was input into the PatCount pattern file, the frequency of "*absolute*" in all the texts was counted by PatCount. Another example is "*noun-noun phrase*". Its regular expression was written as $\S+\_NN\s\S+\_NN\s$. When this pattern file was put into the PatCount software, the frequency count of the *noun-noun phrase* structure in all the texts was automatically performed by PatCount. In this way, the frequencies of all the academic linguistic features in all the texts can be counted.

**3.2.2 Conducting a factor analysis.** Finally, the frequencies of linguistic features obtained from MAT and PatCount were normalized to rates per 1000 words and were put in an Excel file for the next step of factor analysis using SPSS v.25 (see S1 Table). Several pilot factor analyses were performed and only factors with communities over 0.25 and factor loadings higher than 0.30 were retained for final factor analysis. Hence, a total of 62 linguistic features were ultimately retained (see S4 Table). The final factor analysis was run via Principal Axis Factoring with Promax rotation to allow some correlation between factors. A four-factor solution was found to be optimal. The eigenvalues for the four-factor solution accounted for approximately 60.419% of the cumulative variance explained (see S4 Table). Dimension scores for each discipline (see S3 Table) were calculated using z-scores [8] (see S2 Table) to quantify the extent to which a text utilizes the patterns of variation identified by each factor. The mean dimension scores of 100 texts in each discipline of each corpus constituted the dimension score for each discipline of the CRC and NRC for comparison (see Table 2).

ANOVA tests were employed to examine whether there were significant differences between the NRC and CRC along each dimension. With two independent variables (i.e., language background and discipline), this study examined the effects of language background (native English researchers and Chinese researchers) and 12 disciplines (agriculture, art, economics, history, literature, law, medicine, natural science, engineering, education, philosophy and management science) on the dimension score of the newly developed MDA. Due to the non-normal distribution of the data, a nonparametric approach, an Aligned Rank Transform (ART ANOVA) was employed.

**3.2.3 Reliability and validity.** To ensure the validity of this research, we controlled the bias in collecting and analyzing the qualitative data by using the method of triangulation (of investigators); that is, we assigned two research assistants

**Table 2. Mean scores of each discipline on four dimensions in NRC and CRC.**

| Discipline | Dimension | NRC | | CRC | |
|---|---|---|---|---|---|
| | | Mean | SD | Mean | SD |
| 1. Agriculture | D1 | 8.0 | 17.3 | −1.2 | 8.3 |
| | D2 | 3.4 | 16.3 | −0.1 | 9.3 |
| | D3 | −1.4 | 16.8 | −0.1 | 9.5 |
| | D4 | 2.7 | 13.7 | −0.1 | 9.6 |
| 2. Art | D1 | 4.4 | 13.5 | −1.8 | 8.9 |
| | D2 | 2.2 | 13.0 | −0.1 | 8.6 |
| | D3 | −1.0 | 13.4 | 0.7 | 8.4 |
| | D4 | 2.4 | 11.8 | 1.0 | 9.2 |
| 3. Economics | D1 | 4.3 | 12.3 | −2.4 | 8.5 |
| | D2 | 2.6 | 12.1 | 0.3 | 9.3 |
| | D3 | −0.2 | 12.2 | −0.9 | 8.2 |
| | D4 | 2.1 | 11.0 | −0.6 | 8.1 |
| 4. Engineering | D1 | −1.1 | 8.4 | −3.9 | 4.4 |
| | D2 | 2.6 | 10.0 | −2.8 | 5.5 |
| | D3 | 0.0 | 9.2 | 0.3 | 6.7 |
| | D4 | 2.1 | 8.6 | −2.1 | 9.8 |
| 5. History | D1 | 2.1 | 11.4 | −1.2 | 8.0 |
| | D2 | 0.3 | 11.4 | −1.6 | 8.7 |
| | D3 | −2.1 | 11.7 | 0.7 | 9.0 |
| | D4 | 0.5 | 11.0 | −0.4 | 9.3 |
| 6. Law | D1 | 3.6 | 12.6 | −1.5 | 7.7 |
| | D2 | 2.0 | 12.2 | −1.0 | 8.7 |
| | D3 | −0.4 | 12.7 | 0.6 | 8.5 |
| | D4 | 1.8 | 11.7 | −0.4 | 9.7 |
| 7. Literature | D1 | 1.9 | 10.6 | −1.6 | 7.0 |
| | D2 | 0.4 | 10.8 | −0.3 | 8.1 |
| | D3 | −1.23 | 11.0 | 0.5 | 8.4 |
| | D4 | 0.5 | 10.3 | −0.4 | 9.1 |
| 8. Management science | D1 | 0.2 | 9.4 | −2.0 | 7.8 |
| | D2 | −0.1 | 9.8 | −1.2 | 8.7 |
| | D3 | −1.2 | 9.7 | 1.2 | 8.6 |
| | D4 | 0.3 | 10.4 | −1.5 | 8.8 |
| 9. Medicine | D1 | 1.4 | 9.7 | −2.8 | 7.9 |
| | D2 | 1.3 | 9.9 | −1.1 | 8.1 |
| | D3 | 0.8 | 10.9 | 1.4 | 8.1 |
| | D4 | −0.6 | 9.8 | −1.8 | 8.7 |
| 10. Natural science | D1 | 2.0 | 9.9 | −3.2 | 8.1 |
| | D2 | 0.2 | 9.7 | −0.4 | 7.5 |
| | D3 | 0.9 | 9.5 | 0.7 | 8.5 |
| | D4 | 0.6 | 9.7 | −1.5 | 9.0 |
| 11. Education | D1 | 0.3 | 9.4 | −1.9 | 7.4 |
| | D2 | −1.3 | 8.8 | −1.9 | 8.8 |
| | D3 | −0.2 | 9.1 | −0.5 | 7.8 |
| | D4 | −0.5 | 9.8 | −2.7 | 8.5 |

*(Continued)*

**Table 2.** (Continued)

| Discipline | Dimension | NRC | | CRC | |
|---|---|---|---|---|---|
| | | Mean | SD | Mean | SD |
| 12. Philosophy | D1 | 0.0 | 8.3 | −3.5 | 5.5 |
| | D2 | −1.2 | 9.7 | −2.3 | 7.4 |
| | D3 | −0.3 | 8.7 | 1.7 | 7.6 |
| | D4 | 1.1 | 9.4 | −2.6 | 9.6 |

to collect, clean and analyze the data. We trained these research assistants first and ensured that they were completely clear about the criteria of selecting journals and research articles and how to clean the textual data and use the software to analyze the data.

After counting the frequencies of the linguistic features via the computer programs, we obtained an Excel data file (See S1 Table) and then we used the Kaiser-Meyer-Olkin measure of sampling adequacy (KMO) to test the validity of these data before we used SPSS to perform the factor analysis. The KMO value was .974, indicating that the data were suitable for factor analysis (See S4 Table).

## 4. Results and discussion

### 4.1 Results of factor analysis

The KMO of the data in this study yielded a score of .974, indicating that correlation patterns were noticeable in the data [57]. Barlett's Test for Sphericity (Approximate Chi-Square = 198364.098, df = 2628, p = .000) was significant, indicating that adequate correlations existed in the correlation matrix and that factor analysis was suitable for the data in this study [58]. After conducting factor analysis, four dimensions were extracted and interpreted in the following sections.

### 4.2 Interpretation of the dimensions

Four dimensions were extracted through factor analysis. Following Biber's [8] way of interpreting the dimensions, the detailed analysis and discussion of each dimension is presented in the following sub-sections.

**4.2.1 Dimension one: Academic involvement and interaction versus information density.** Dimension one, labelled as Academic Involvement and Interaction versus Information Density, is composed of sixteen positive features and three negative features (see Table 3).

On this dimension, positive linguistic features (e.g., first-person pronouns, reporting verbs, public verbs, attitude markers, necessity modals, emphatics, and boosters) are mostly associated with elaborate and involved language production. First-person pronouns are markers of ego-involvement in a text [8]. They are used to bring author and readers into the discourse, creating a sense of interaction. Attitude markers, emphatics and boosters are devices for self-mentioning and expressing one's attitude in language use, a textual signal by which the writers personally involve themselves in their discourse to interact with the readers. These features create a picture of involving the author's interaction with the readers to highlight the author's stance. On the other hand, the features loaded negatively on Dimension 1 are all noun-phrase structures (nouns, nominalizations and prepositional phrases), closely aligned with Biber's [8,35] and Gray's [59] findings in which they associated these features with an informational purpose. This utilization of structures is meant to pack high amounts of information into academic nominals [8]. Overall, for Dimension 1, positive features exhibit a more academically involved and interactional style, whereas negative features reveal a more information density writing style.

**Table 3. Linguistic features on Dimension 1.**

| Positive linguistic features with loadings | | | |
|---|---|---|---|
| Total adverbs (.953) | Infinitives *TO* (.948) | Emphatics (.913) | Reporting verbs (.905) |
| Pronoun *IT* (.900) | Present tense (.892) | Attributive adjectives (.886) | Perfect aspect (.874) |
| Attitude markers (.870) | Public verbs (.844) | Subordinator *that* deletion (.844) | WH- clause (.826) |
| Boosters (.813) | Identity markers (.797) | First person pronoun (.793) | Necessity modals (.726) |
| **Negative linguistic features with loadings** | | | |
| Total other nouns (−.934) | Total prepositional phrases (−.789) | Nominalizations (−.551) | |

Fig 1 shows the dimension scores across the 12 disciplines and language background on Dimension 1. It can be seen that native English researchers' academic writing is more involved and interactional than that of Chinese researchers. ART ANOVA tests were first used to examine the overall main effect of language background, discipline and the interaction between them on Dimension 1. The results reveal that the main effect of language background on Dimension 1 is statistically significant ($F = 75.940$, $p < 0.001$, partial eta-squared = 0.031), suggesting that Dimension 1 can differentiate research articles written by native English researchers and Chinese researchers. Given that the overall main effect of language background on Dimension 1 was significant, we further conducted Dunn's post-hoc pairwise comparisons and applied the Benjamini-Hochberg (BH) method to adjust the p-values for controlling the false discovery rate to confirm on what disciplines researchers with different language backgrounds differed from each other. Dunn's test suggests that the difference between native English researchers' academic writing and Chinese researchers' is significant for the disciplines of agriculture ($p = 0.004$), art ($p = 0.008$), economics ($p = 0.002$), law ($p = 0.034$), medicine ($p = 0.016$), natural science ($p = 0.004$) and philosophy ($p = 0.029$), but insignificant for the disciplines of history, literature, engineering, education and management science, as shown in Fig 1.

Moreover, ART ANOVA tests reveal that the overall main effect of discipline ($F = 2.42$, $p = 0.006$, partial eta-squared = 0.011) is statistically significant, but the interaction effect between language background and discipline ($F = 1.190$, $p = .0.292$, partial eta-squared = 0.006) is insignificant, indicating that disciplines influence Dimension 1 scores but the interaction between language background and discipline has no effect on Dimension 1 scores. Dunn's post-hoc pairwise comparisons with BH method was then done to identify the disciplinary differences within the same language background because the overall main effect of discipline on Dimension 1 scores is significant. Dunn's test shows that the differences between disciplines within Chinese researchers' academic writing are insignificant on Dimension 1, but in native researchers' academic writing, the differences between the disciplines of agriculture and education ($p = 0.032$), agriculture and engineering ($p = 0.004$), agriculture and management science ($p = 0.018$), agriculture and philosophy ($p = 0.027$), art and engineering ($p = 0.029$), economics and engineering ($p = 0.013$), economics and management science ($p = 0.050$) are significant, as shown in Fig 2.

Two examples were excerpted from the agriculture discipline from native English researchers corpus and engineering discipline from Chinese researchers corpus because the dimension scores of these two disciplines are the highest on the positive and negative ends, respectively. In the two examples, positive features are marked in boldface and negative features are underlined (this procedure is applied through all 8 typical examples). In Example (1), the researcher heavily relies on the use of positive features, such as first-person pronouns (e.g., *we, our*), reporting verbs (e.g., *require, propose*), adverbs (e.g., *directly, instead, wholly, surprisingly, conceptually*), attributive adjectives (*previous, multiple, simple, significant*) and present tense to highly stress and recommend their novel and effective research approach to the reader.

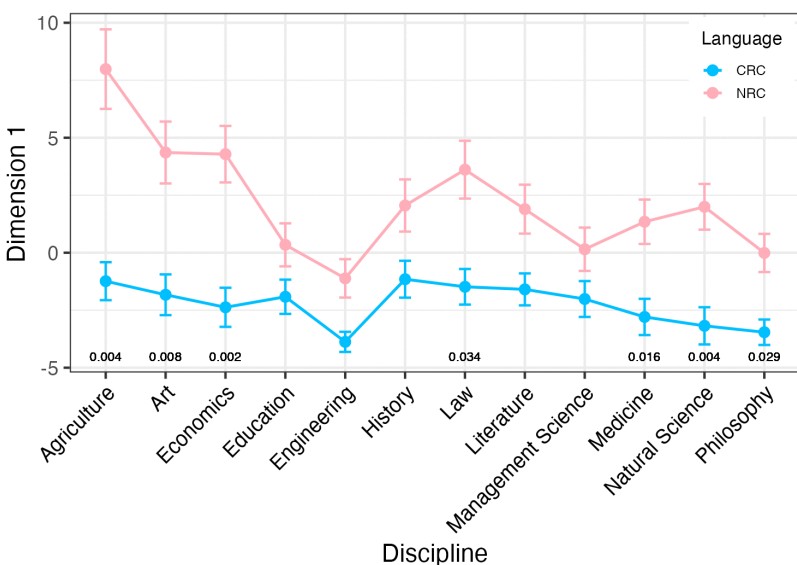

**Fig 1. Dimension 1 scores across disciplines and language background.**

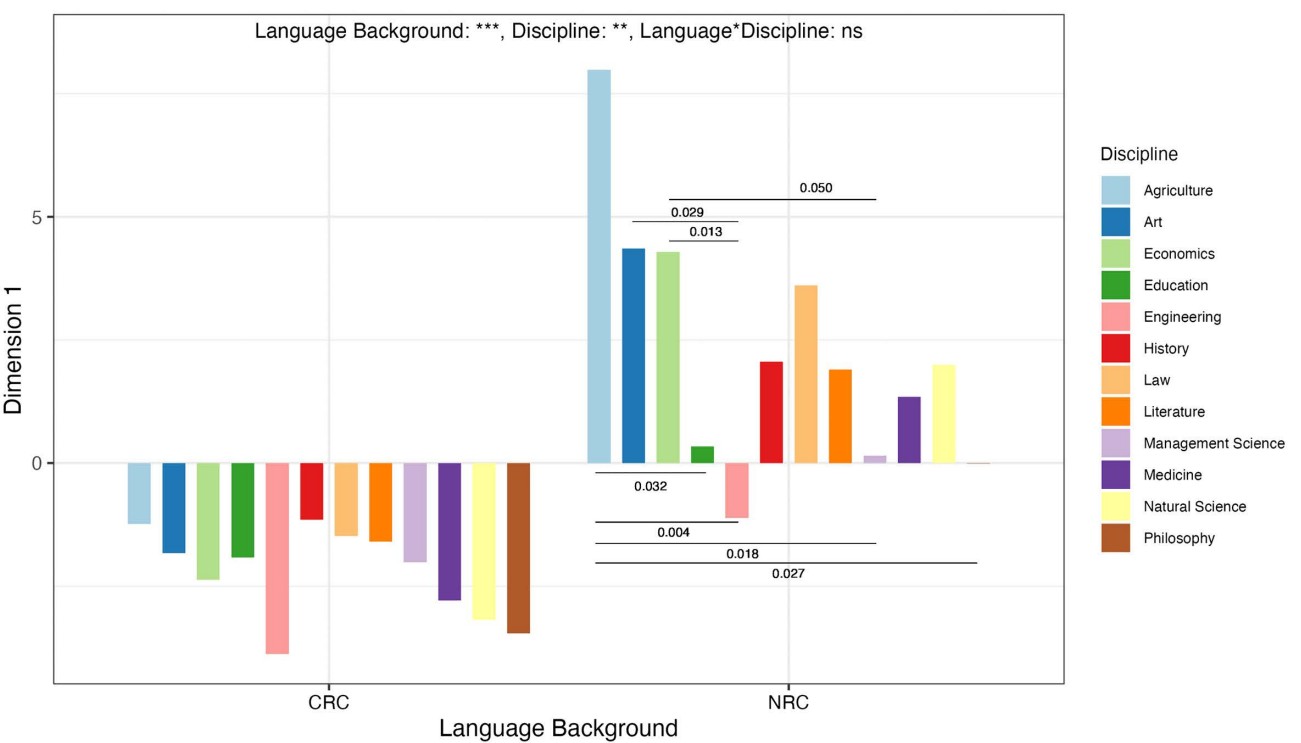

**Fig 2. Disciplinary differences within the same language background on Dimension 1.**

Choosing these linguistic features reveals that the researcher would like to explicitly convey their stance to persuade readers about their innovation reported in their research. There are very few negative linguistic features, such as nominalization and preposition, and only some necessary nouns are found in Example (1).

(1) As far as **we** know, **our** approach **is** the first to generate bounding box proposals **directly** from edges. Unlike all **previous** approaches **we do** not use segmentations or super pixels, nor **do we require** learning a scoring function from **multiple** cues. **Instead**, **we propose** to score candidate boxes based on the number of contours **wholly** enclosed by a bounding box. **Surprisingly**, this **conceptually simple** approach **out-competes previous** methods by a **significant** margin.

(NRC_AGRI27)

In contrast, Example (2) shows the researchers' preference for negative linguistic features on Dimension 1. Using a large number of nouns, nominalizations and prepositions mark the article with dense information. Compared with Example 1, Example 2 avoids involving and interacting with the readers to conceal their authorial identity and opinion by using rare positive features on Dimension 1, but it is featured with conveying information objectively.

(2) Nick and Satao compared the aerodynamic <u>characteristics of</u> two <u>models</u>—short and optimized—to determine the best <u>pod shape for drag reduction</u>. The <u>pressure drag of</u> the optimized <u>model</u> was 24% less than that <u>of</u> the short <u>model</u>.

(CRC_ENGR01)

Dimension 1 in this study reflected a distinction that has been consistently identified in MD analyses, i.e., it was more involved language production and high-density informational language. In the academic writing register, native English researchers were more likely to involve themselves and interact with readers to clearly and confidently express their opinions than Chinese researchers were. This finding aligned with those of previous studies. Native English writers preferred to project themselves in explicit interactional academic discourse, construct authorial presence and show their engagement with their readers in arguments [60]. While EFL learners were overwhelmingly reluctant to establish an authorial identity in their writing, Chinese EFL learners presented a passive, test-oriented writer identity due to influences from learning and testing cultures [61]. Similar findings were reported by Liu and Zhang [62], who found that the frequency of interactional metadiscourse utilized by Chinese master students in academic writing was less than that used by international journal authors. This was perhaps because that Chinese researchers, as EFL learners, were taught to eliminate explicit agency in their academic writing [63] to establish objectivity. Chinese students were taught that academic writing was a rigorous writing so that the authors should avoid involving or engaging the readers in the discourse and not claim their authorial presence. Therefore, Chinese researchers preferred to disguise their interpretative statements for objectivity, whereas Anglo-American rhetoric tended to project a more reader-oriented attitude characterized by more reader guidance and an explicit authorial presence [64]. While native researchers excel at authorial engagement and elaboration, Chinese researchers demonstrate strengths in information density and conciseness, efficiently packing content with minimal redundancy, a style valued in fast-paced, technical fields as engineering. This aligns with English for Academic Purposes (EAP) research showing L2 writers' precision can model concise argumentation for L1 peers. Thus, both groups offer mutual lessons of natives in interactivity and Chinese in streamlined reporting.

This distinction on Dimension 1 was especially significant on the disciplines of agriculture, art, economics, law, medicine, natural science and philosophy, but insignificant in the disciplines of education, engineering, history, literature and management science, meaning that native researchers prefer a more involved and interactional writing style while

Chinese researchers tend to be informational in the disciplines of agriculture, art, economics, law, medicine, natural science and philosophy. These disciplines belong to hard sciences, except art, law and philosophy based on the existing body of disciplinary classification theories [65–67]. Although art, law and philosophy disciplines do not belong to hard sciences, they are regarded as pure sciences compared with applied sciences. Pure sciences are concerned with the development of knowledge for its own sake and testing theoretical formulations [66], rather than with any form of practical outcome [68]. Therefore, native English and Chinese researchers show significant differences in writing in hard or pure sciences with the characteristics of having highly internationalized research paradigms, rigorous reporting standards and a relatively unified terminological system. Interestingly, it was found that even for hard sciences that emphasize objectivity, logical rigor and highly conventionalized formats, native researchers exhibit more flexibility by appropriately incorporating authorial identity and engagement with the readers, rather than being informational and objective as it should be. This finding demonstrated native researchers strategically declared their author stance within the disciplinary norms due to their greater writing fluency and more liberal expressions in writing. Whereas, Chinese researchers appeared to be conservative in writing in hard and pure disciplines, by strictly adhering to hard sciences' disciplinary conventions, focusing on objective facts and rigorous academic norms. In Hyland's study [69], he pointed out that the conventions of the hard sciences stress the demonstration of objectivity, precision, and logical reasoning, minimizing the researcher's role to highlight the phenomena under study, while in the soft disciplines, the interpretation of personal experiences is central and writers are more able to draw on their own authority to persuade readers. Our finding aligns with Hyland's [69] observation in disciplinary variation in his meta-discourse theory, but extends his theory by finding out that the disciplinary variation between hard sciences and soft sciences was also influenced by second language writing challenges and language proficiency (i.e., competence). Native researchers' writing style in hard science is not completely informative and objective as expected; on the contrary, they also showed an involved and interactional writing style. However, Chinese researchers, as L2 writers, often adhere rigidly to impersonal constructions in hard science. This rigidity may stem from high cognitive load [70] during composition in English, leaving limited capacity for rhetorical flexibility. Conversely, native researchers' linguistic proficiency affords them the flexibility to strategically follow these conventions. As Hyland [71] suggested, expert writers employ self-mention not to violate objectivity, but to skillfully construct their authorial credibility and responsibility within the disciplinary framework. Thus, in hard or pure sciences, native researchers are more strategic to freely express their author stance, but Chinese researchers tend to follow the generally pre-set disciplinary conventions.

In the disciplines of education, engineering, history, literature and management science, which are generally regarded as soft disciplines except engineering, the difference between native researchers being involved and interactional and Chinese researchers being informational is insignificant. Soft disciplines and engineering discipline falling under applied sciences, are characterized by rhetorical and personalized nature emphasizing personal interpretation more where the individual voice and argumentative skill of the researcher are paramount [68]. Therefore, academic writing in soft disciplines permits a diversity of rhetorical styles and modes of expression. This intrinsic variability may obscure differences induced by various language backgrounds, making statistical differences insignificant.

Fig 2 shows the disciplinary differences within the same language background. It can be seen that within Chinese researchers the variations among disciplines are insignificant, but within native researchers, the differences between agriculture and education, agriculture and engineering, agriculture and management science, agriculture and philosophy, art and engineering, economics and engineering, economics and management science were significant. All disciplines except engineering within native researchers' language background located on the positive end of Dimension 1 (dimension score is more than 0), meaning that these 11 disciplines exhibit a generally involved and interactional writing style (Fig 2). Statistically significant differences observed between agriculture and education, agriculture and management science, agriculture and philosophy, economics and management science were only in the extent to which they all exhibit the features of being involved and interactional. Notably, among all disciplines, only engineering stood out on the negative end of Dimension 1 (dimension score is below 0), meaning that native

researchers preferred a writing style of being informational in engineering discipline. Engineering discipline was significantly different from the soft sciences as art and economics and other hard applied sciences as agriculture, as shown in  Fig 2. This phenomenon can be explained by the subtle distinctions among disciplines. Engineering is part of applied sciences that relies heavily on mathematical equations to design or create new devices or structures. Its emphasis on precision, predictability, and safety necessitates a highly anonymized, manual-like writing style to convey technical information, more than personal interpretation and rhetorical expression [68]. In contrast, the research topics of agriculture science are often complex living systems, economics deals with the production, distribution, and consumption of goods and services, and art is grounded in personal aesthetic interpretation. These three disciplines, while also providing information, must argue for and interpret the contextual and uncertain nature of their findings [72]. Consequently, their writing requires authorial intervention and interactive elements, thereby exhibiting involved and interactional writing style. This finding powerfully demonstrates that the influence of disciplines on writing style is deeply embedded within each discipline, beyond a simple hard-soft binary.

**4.2.2 Dimension two: Interactive argumentation versus static description.** Dimension two is made up of thirteen positive features and six negative features (see Table 4), labelled as Interactive Argumentation versus Static Description.

Linguistic features such as transitional signals, conjuncts, participles, adverbial clauses and relative clauses co-occurred on positive end of Dimension 2, providing an interactive function of managing information flow to establish writers' interpretations. Conjuncts explicitly mark logical relations between clauses; and participles are used for integration or structural elaboration [8]. Adverbial clauses and relative clauses serve the function of cohesion by connecting clauses in a good logic. Therefore, the positive features for Dimension 2 help create this interactive argumentation discourse, showing that in the academic writing register, the author is aware of the logical relations between clauses and paragraphs to form argumentative discourse.

Fig 3 shows that native English researchers' academic writing is more interactive and argumentative than that of Chinese researchers. ART ANOVA tests suggest that the main effect of language background on Dimension 2 is statistically significant ($F = 12.685$, $p = 0.000$, partial eta-squared $= 0.005$). Since the overall main effect of language background on Dimension 1 was significant, Dunn's post-hoc pairwise comparisons were conducted to confirm what disciplines the researchers with different language backgrounds differed from each other. Dunn's test shows that the difference between native English researchers' academic writing and Chinese researchers' is statistically significant only in the discipline of engineering ($p = 0.010$). ART ANOVA tests also reveal that the main effect of both discipline ($F = 1.581$, $p = 0.098$, partial eta-squared $= 0.007$) and the interaction effect between language background and discipline ($F = 1.024$, $p = 0.422$, partial eta-squared $= 0.005$) on Dimension 2 scores are not statistically significant.

Examples (3) and (4) were chosen from the native researchers' agriculture discipline and the Chinese researchers' engineering discipline, respectively representing texts with the highest positive dimension scores and the highest negative dimension scores on Dimension 2.

In Example (3), the researchers argue an event using interactive devices through positive features on Dimension 2 (i.e., *conjuncts, phrasal coordination, present participial WHIZ deletion relatives, past participial WHIZ deletion relatives, adverbial subordinators,* and *concessive adverbial subordinators*). By using conjuncts and participial structures, cohesion is achieved. Researchers present a clear relationship between clauses and sentences. These positive features provide a picture of interactive argumentation.

(3)   **Although** the amyloid load reached a plateau early after symptom onset, astrocytosis **and** microgliosis **increased** linearly throughout the disease course, **thus integrating** both lesions as a marker of disease severity. **Moreover**, glial responses **correlated** positively with tangle burden, **whereas** astrocytosis **correlated** negatively with cortical thickness. **However**, neither correlated with amyloid load.

(NRC_AGRI79)

**Table 4. Linguistic features on Dimension 2.**

| Positive linguistic features with loadings | | | |
|---|---|---|---|
| Transitional signals (.975) | Present participial clauses (.905) | Conjuncts (.878) | Phrasal coordination (.877) |
| Other adverbial subordinators (.867) | Past participial clauses (.862) | Split auxiliaries (.853) | Present participial WHIZ deletion relatives (.826) |
| Concessive adverbial subordinators (.820) | Pied-piping relative clauses (.815) | Past participial WHIZ deletion relatives (.808) | Causative adverbial subordinators (.715) |
| Conditional adverbial subordinators (.662) | | | |
| **Negative linguistic features with loadings** | | | |
| Existential *there* (−.861) | *Be* as main verb (−.826) | Demonstrative pronouns (−.814) | Third person pronouns (−.790) |
| Past tense (−.783) | Predicative adjectives (−.779) | | |

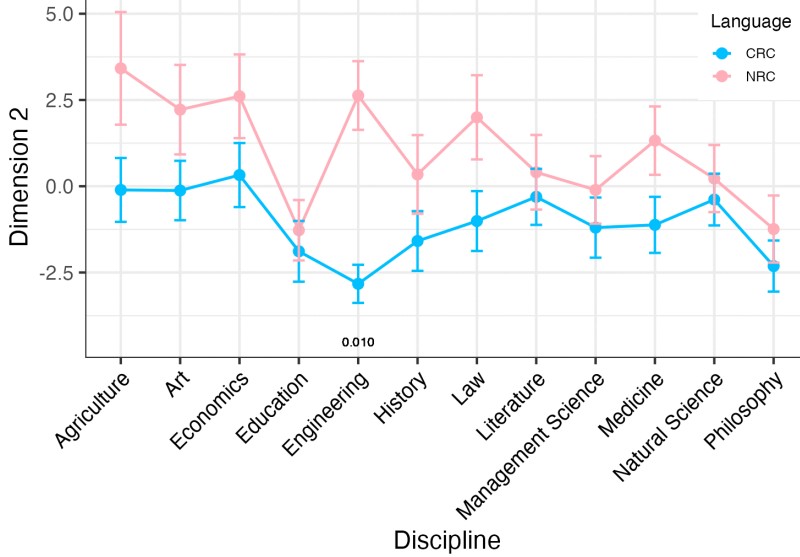

**Fig 3. Dimension 2 scores across disciplines and language background.**

On the contrary, Example (4) exhibits the function of negative linguistic features, creating a static description discourse. The *BE* verb, existential there structure and *predicative adjective* are markers of the static, informational style that are common in writing, but they are considered non-complex constructions with a reduced informational load, thus being informal as a spoken style [8]. Unlike active verbs, *BE* and existential there are usually used to describe a static state. In Example (4), simple sentence construction of "Subject + be" is used three times in a paragraph to describe a simple and basic fact about democracy.

(4) Democracy <u>is</u> not a simple form of government and judgments about the nature of different governments that claim to <u>be democratic</u> should not be made in a simplistic manner. Certainly, degrees of democracy <u>are possible</u> and a crucial criterion <u>is</u> the proportion of citizens voting correctly at any particular time.

(CRC_ENGR38)

In summary, Dimension 2 differentiated native English researchers' interactive argumentation writing style from Chinese researchers' static descriptive writing style. This was explainable because previous studies have shown that native English researchers tended to present a greater number of logical markers than L2 writers do [73,74]. To achieve logical elaboration, a good command and proficiency in the English language are needed. Chinese researchers, as EFL learners, usually presented a less causal content than the native speaker's writing [75] and inadequate use of logical connectors [76]. Native researchers exhibited an explicit logical connection of the ideas in a linear way, but Chinese researchers did not.

On the other hand, Chinese researchers presented a more static descriptive writing style. EFL learners tended to use *is* and *are* frequently, and usually added a *be* verb into a simple subject structure to express meaning [77]. Chinese researchers' English vocabulary was not rich enough, and they cannot organize and combine various language bundles skillfully; thus, their sentence structures were mostly simple repetitions in collocation, using less complex sentence patterns to enrich and refine meaning expression. The same held for existential *there*, Chinese university students' excessive use of existential sentences in their academic writing was reported by Zhang et al. [78] Dimension 2 differentiated native researchers' interactive argumentation from Chinese researchers' static descriptive style. This reflects proficiency differences that natives chain ideas linearly with complex connectors, while Chinese EFL writers favor direct *A is B* structures due to vocabulary and syntax constraints. Yet this L2 simplicity confers strengths in clarity and conciseness, avoiding native overelaboration that may obscure meaning. Such straightforwardness enhances readability in disciplines with heavy jargons.

In the discipline of engineering, this difference was statistically significant. Writing in engineering emphasizes preciseness and objectivity to convey the information, usually complex profession-related terms and technical information. L2 Chinese researchers, likely constrained by their limited language proficiency and the cognitive load of writing in a second language, tend to produce texts with simple sentence structures in such a discipline requiring professional information and demanding logical rigor. In contrast, native researchers, proficient in their discourse community, can strategically employ interactive devices to guide readers through reasoning, not limited to static description, in a hard discipline.

**4.2.3 Dimension three: Impersonal evaluation versus personal opinion.** Interpreted as an Impersonal Evaluation versus Personal Opinion, Dimension 3 is characterized by eight positive features and six negative features (see Table 5). The use of passives provides a sense of objective detachment in expository prose. This sense of objectivity is part of scientific culture, and is often expected in scientific writing [35]. Another two features (downtowners and hedges) generally reflect the uncertainty of the language. Positive features show the author's preference to be impersonal in presenting an objective fact without expressing their subjective opinion, whereas negative features indicate personal and persuasive discourse.

Fig 4 shows the distribution of the mean dimension scores across the 12 disciplines and language background along Dimension 3. Chinese researchers' academic writing is generally more impersonal than that of the native English researchers in all disciplines except economics, natural science and education, whereas native English researchers' writing is all at the negative end of Dimension 3 except for medicine, natural science and engineering. ART ANOVA tests

**Table 5. Linguistic features on Dimension 3.**

| Positive linguistic features with loadings | | | |
|---|---|---|---|
| Agentless passives (.865) | *By*-passives (.805) | Noun-noun phrase (.792) | Hedges (.762) |
| Demonstratives (.744) | Possibility modals (.744) | Downtoners (.704) | Analytic negation (.617) |
| Negative linguistic features with loadings | | | |
| Suasive verbs (−.974) | *That* relative clauses on object position (−.947) | Amplifiers (−.940) | *That adjective complements (−.932)* |
| *That* verb complements (−.887) | Predicative modals (−.860) | | |

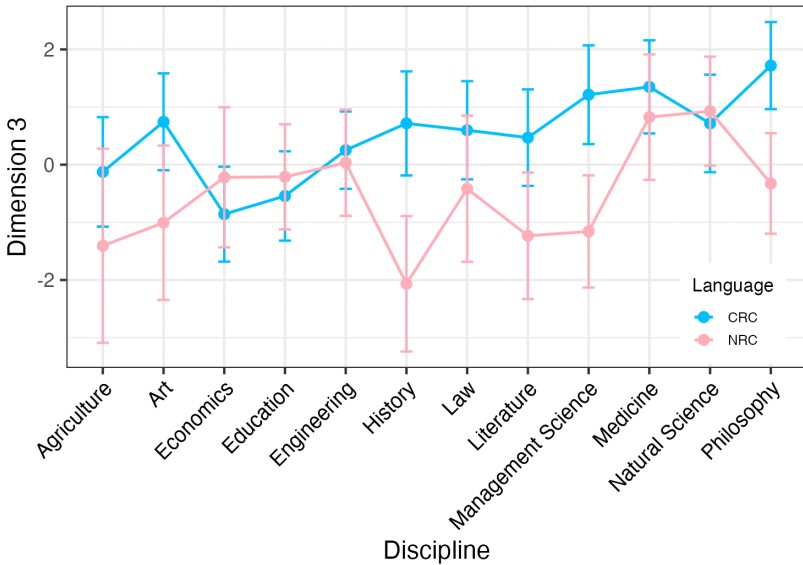

**Fig 4. Dimension 3 scores across disciplines and language background.**

reveal that the main effect of language background on Dimension 3 is statistically significant ($F = 10.330$, $p = 0.001$, partial eta-squared = 0.004), suggesting that Dimension 3 could differentiate research articles written by native English and Chinese researchers. But Dunn's post-hoc pairwise comparisons with BH method was then done to find no statistically significant difference for any discipline. Meanwhile, the main effect of discipline ($F = 1.010$, $p = 0.435$, partial eta-squared = 0.005) and the interaction effects between language background and discipline ($F = 0.730$, $p = 0.711$, partial eta-squared = 0.003) on Dimension 3 are not statistically significant.

Example (5) was selected from the Chinese researcher corpus in the philosophy discipline because it had the highest positive dimension scores. Researchers are showing their impersonal tone when presenting human research ethics by using passive structures, demonstratives and noun-noun phrases.

(5)  Human **Research Ethics** aims to ensure that research **is conducted** to the highest ethical standard and **that** human participants in research **are protected**.

(CRC_PHIL05)

In contrast, negative features on Dimension 3 suggest a personal opinion expressing style in writing. Example (6) was selected from the native researchers' corpus in the discipline of history because it had the highest negative dimension score. Suasive verbs imply intentions to make some change in the future (e.g., command, stipulate) [8] and amplifiers have the effect of boosting the force of the verb [79]. In Example (6), suasive verbs, *that* verb complements and *that* relative clauses in the object position are used when the researchers are making conclusions for their research. This reveals that the researchers are stating their personal opinions and trying to persuade the readers about their findings.

(6)  We <u>insist that</u> the prefixal agreement and the theme suffix are fundamentally different kinds of morphemes. Therefore, we <u>propose that</u> the special character of inverse contexts arises from the fact that the EA never agrees with the core probe, and this failure is what must be repaired.

(NRC_HIST95)

Dimension 3 distinguished the native English researchers' and the Chinese researchers' writing in that the Chinese researchers generally adopted a more impersonal style, whereas the native researchers tended to make personal argumentation in their research. This finding aligned with previous research in which the frequent use of simple passive verbs was found in L2 learners' academic writing [80]. Chinese impersonality offers strengths in objectivity and humility, aligning with disciplinary conventions that prioritize collective knowledge over individual voice. This restrained writing style models with cultural sensitivity for natives who risk perceived bias through overt self-positioning.

**4.2.4 Dimension four: Explicit elaborating style versus simplified reporting style.** Six positive features and four negative features load on Dimension 4 (see Table 6) which is labelled as Explicit Elaborating Style versus Simplified Reporting Style.

Positive features on Dimension 4 create a picture of explicit elaboration. Explicitness has been measured by features such as word length and the type/token ratio which is the ratio of the number of different words to the total number of words [8]. They are features revealing lexical specificity. Complex words and a careful selection of vocabulary result in a high type/token ratio [8], indicating good syntactic complexity. Together with relative clause structures, these features reflecting authors' proficiency in the English language serve the function of an explicit explanation of the authors' argumentation.

Fig 5 shows the distribution of the mean scores of Dimension 4 across disciplines and language background. The native English researchers use a more explicit elaborating style than the Chinese researchers do in academic writing. ART ANOVA tests reveal that the main effect of language background ($F = 14.210$, $p = 0.000$, partial eta-squared = 0.006) on Dimension 4 are statistically significant. Then Dunn's post-hoc pairwise comparisons with BH method was conducted, this difference was not statistically significant for any discipline. Meanwhile, the main effect of discipline ($F = 1.380$, $p = 0.177$, partial eta-squared = 0.006) and the interaction effect between language background and discipline on Dimension 4 are insignificant ($F = 0.560$, $p = 0.863$, partial eta-squared = 0.003).

Examples (7) and (8) were chosen from disciplines with the highest positive and negative dimension scores, respectively on Dimension 4. In Example (7), positive features on Dimension 4 as split infinitives, *that* relative clauses in the subject position and independent clause coordination are used to help explicitly illustrate the benefits of sustainable farming. (The type/token ratio is computed by counting the number of different lexical items, so it is not marked in Example 7).

(7)  Sustainable farming practices, **that** aim **to significantly improve** soil health and biodiversity, **not only help to reduce environmental impact but also strive to increase crop yields**.

(NRC_AGRI09)

In contrast, the cooccurrence of negative features such as private verbs, gerunds, contractions, and relative clauses on Dimension 4 suggests that researchers prefer simple structures to report their research more colloquially. Private verbs [62] describe or refer to mental states (e.g., know, learn, think) and non-observable intellectual acts that are private, such

**Table 6. Linguistic features on Dimension 4.**

| Positive linguistic features with loadings | | |
| --- | --- | --- |
| Type-token ratio (.947) | Independent clause coordination (.932) | Sentence relatives (.872) |
| Word length (.840) | *That* relative clauses on subject position (.769) | Split infinitives (.628) |
| **Negative linguistic features with loadings** | | |
| WH relative clauses on subject position (−.900) | Contractions (−.850) | Private verbs (−.788) |
| Gerunds (−.704) | | |

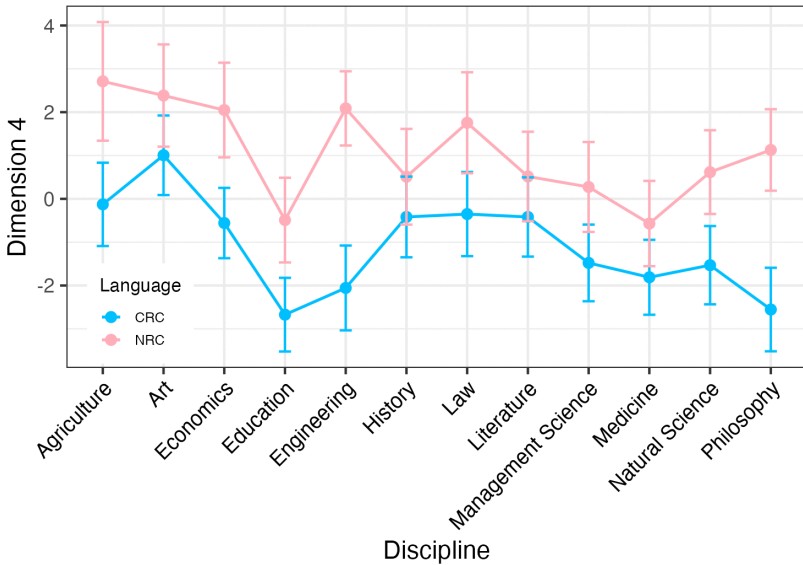

**Fig 5. Dimension 4 scores across disciplines and language background.**

as emotive acts (feel, hope), mental acts (realize, understand), and cognitive acts (believe, conclude, forget, recognize) [81]. In Example (8), the researchers focus more on reporting their personal opinions when using private verbs.

(8) Moreover, the authors <u>hope</u> to improve the prediction results by <u>tuning</u> the hyperparameters and <u>designing</u> more sophisticated features using deep learning models.

(CRC_EDU46)

In summary, native English researchers preferred a more explicit elaborating style because native English researchers had much better language proficiency than Chinese researchers did. They certainly surpassed Chinese researchers in terms of syntactic complexity, lexical diversity and explicit elaboration. Chinese researchers were not skilled English language users and most of them could use only simple structures to report their research in their writing. Dimension 4 contrasted native researchers' explicit elaborating style with Chinese researchers' simplified reporting style. Natives excelled in elaboration due to superior proficiency. Chinese simplicity provides strengths in efficiency and readability, prioritizing research content over linguistic display. This reader-centered writing style models concise impact for natives prone to overelaboration.

## 5. Limitations

While this study offers novel insights into L1/L2 academic writing patterns, two methodological limitations warrant consideration. First, author classification proxies have inherent edge cases. Second, disciplinary sampling lacks sub-disciplinary nuance explained further below.

### 5.1 Author classification limitations

Although our double-safeguard method, i.e., institutional restriction to US/UK/Canada/Australia/NZ with surname origin matching, virtually eliminates L2 misclassification as L1, rare edge cases persist (e.g., naturalized citizens with fully

Westernized names). These proxy limitations, while standard in large-scale EAP research, suggest caution in future validation through direct author verification.

## 5.2 Disciplinary sampling limitations

Following China's Level One Disciplines system, we sampled broadly across fields without subdiscipline stratification. This captures major disciplinary tendencies but misses intra-discipline variation, potentially masking specific field patterns. Representativeness is thus sufficient for broad discipline-level effects rather than fine-grained subfield differences. Future studies should build stratified corpora to enhance generalizability across specialized academic subdomains.

## 6. Conclusions

While MDA has effectively captured linguistic variation across general registers and L1/L2 learner corpora, few studies have developed genre-specific MDA frameworks for academic research articles by identifying linguistic features for certain disciplines [35,82]. Existing comparisons in academic writing using MDA method typically focused on L1/L2 variations rather than disciplinary nuances. This study addresses these gaps by: (1) selecting 62 linguistic features for academic writing; (2) using PatCount software for automated tagging; and (3) deriving four novel dimensions that reveal nuanced L1/L2 differences beyond the hard-soft science binary. These contributions provide both a new analytical model and pedagogical tools absent in the previous MDA research.

The current study analyzed the co-occurrence of patterns of the selected 62 linguistic features for academic writing which resulted in constructing a novel multidimensional analysis model with four dimensions distinguishing native English researchers' and Chinese researchers' academic writing differences across twelve disciplines. They are (1) academic involvement and interaction versus information density; (2) interactive argumentation versus static description; (3) impersonal evaluation versus personal opinion; and (4) explicit elaborating style versus simplified reporting style.

ART ANOVA confirmed significant language background effects across all dimensions (p < .001), with discipline effects limited to Dimension 1 and no interaction. Natives showed greater academic involvement, interaction, and elaboration while Chinese researchers favored a more informational, static, descriptive, impersonal and simplified reporting style in their writing.

Native researchers were more academically involved and interactional primarily in agriculture, art, economics, law, medicine, natural science, and philosophy (Dunn's post-hoc, BH-adjusted, p < .05), while Chinese researchers tended to be more informational in these seven disciplines, indicating that in rigorous disciplines as hard or pure sciences, native researchers are more strategic to freely express their author stance, but Chinese researchers tend to be conservative and follow the generally pre-set disciplinary conventions. While these differences were insignificant in soft or applied fields because writing in soft disciplines permits a diversity of rhetorical styles and ways of expression. This intrinsic variability may obscure differences induced by various language backgrounds. Notably, engineering discipline revealed intra-native variation, i.e., natives tended to be more informational than in soft sciences (art & economics) or other hard fields (agriculture), and only in engineering discipline Chinese researchers showed a more static, descriptive and simplified reporting style than native English researchers did, challenging simple hard-soft binaries.

These findings proved that researchers' preference for linguistic features in different disciplines in their academic writing shaped their unique writing styles, and this preference was not simply attributed to hard-soft sciences binary difference, but also related to researchers' language proficiency and subtle distinctions within disciplines under same hard or soft sciences. This suggests that Chinese researchers should exhibit their authorial presence, interact with readers more, and employ more interactive devices to make their writing coherent and explicit.

In addition to the above-mentioned key findings from different language backgrounds and disciplines, this research also contributed to enriching the existing MDA research by providing a newly complementary perspective. Unlike previous studies stressing the differences, this research innovatively emphasized the mutual strengths from these explored

differences. This balanced comparison reveals that mutual strengths of Chinese conciseness (Dimensions 1 & 4) model clarity and efficiency for natives prone to verbosity; native involvement and interaction (Dimensions 1 & 4) guides Chinese toward engagement. Chinese impersonality (Dimension 3) offers objectivity where natives risk bias; natives' logic (Dimension 2) complements L2 simplicity.

This study contributed pedagogically and methodologically as well. Pedagogically, the newly developed model (Fig 6) enables targeted EAP instruction by discipline nuances. Educators should tailor pedagogical approaches to address distinct writing styles between native English and Chinese researchers across various disciplines. By interpreting discipline-specific writing styles on the four dimensions of this novel MDA model, educators can increase students' discipline awareness. Being aware of disciplinary variation and native researchers' writing style empowers learners to engage in quality academic practices. Methodologically, PatCount automation with Regular Expression code writing overcomes traditional MDA tagging barriers, scalable for genre-specific models.

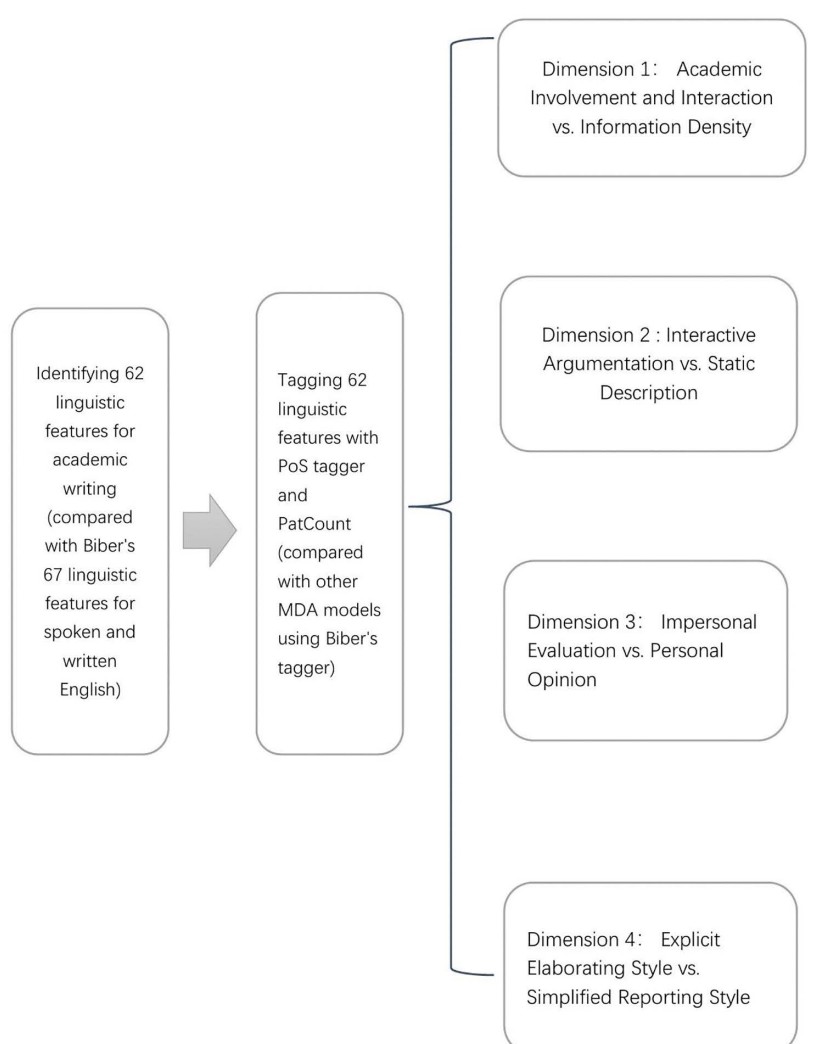

**Fig 6. Newly developed MDA model for academic writing in research articles.**

Limitations in this study include proxy L1/L2 classification and broad disciplinary categories without subdiscipline stratification. Future research should test proficiency subgroups and finer-grained fields.

This framework advances MDA toward academic genres, equipping non-native researchers with discipline-sensitive tools while recognizing cross-cultural writing assets.

## Supporting information

**S1 Table. Frequencies of all linguistic features in 2400 articles for factor analysis.**
(XLSX)

**S2 Table. Z-score of each article on four factors after factor analysis.**
(XLSX)

**S3 Table. Mean dimension score of each discipline on four factors.**
(XLSX)

**S4 Table. Results of factor analysis (KMO, cumulative variance explained and pattern matrix).**
(DOC)

**S5 Appendix. Linguistic features selected in this study.**
(PDF)

**S6 Appendix. List of abbreviations.**
(PDF)

## Acknowledgments

We are grateful to Yang Yang and Jie Liu for their assistance with data collection and data cleaning, to Mei Feng for her insightful comments on the draft research, and to Wanmeng Xiao for her guidance in the data analysis. We also thank the reviewers and editors for providing constructive feedback on the manuscript to improve it.

## Author contributions

**Conceptualization:** Jiaqi Deng, Ghayth Kamel Shaker Al-Shaibani.

**Data curation:** Jiaqi Deng.

**Formal analysis:** Jiaqi Deng, Ghayth Kamel Shaker Al-Shaibani.

**Investigation:** Jiaqi Deng, Ghayth Kamel Shaker Al-Shaibani.

**Methodology:** Jiaqi Deng, Ghayth Kamel Shaker Al-Shaibani.

**Project administration:** Ghayth Kamel Shaker Al-Shaibani.

**Resources:** Jiaqi Deng.

**Software:** Jiaqi Deng.

**Supervision:** Ghayth Kamel Shaker Al-Shaibani.

**Validation:** Jiaqi Deng, Ghayth Kamel Shaker Al-Shaibani.

**Visualization:** Jiaqi Deng.

**Writing – original draft:** Jiaqi Deng.

**Writing – review & editing:** Ghayth Kamel Shaker Al-Shaibani.

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
