## [Decision Letter · Decision Letter 0]

28 Jan 2025

Dear Dr. Al-Shaibani,

Thank you for submitting your manuscript to PLOS ONE. After careful consideration, we feel that it has merit but does not fully meet PLOS ONE’s publication criteria as it currently stands. Therefore, we invite you to submit a revised version of the manuscript that addresses the points raised during the review process.

We look forward to receiving your revised manuscript.

Kind regards,

Fasih Ahmed

Academic Editor

PLOS ONE

Additional Editor Comments:

Dear Author,

The review regarding your article are complete now. The reviewers have suggest the changes. You are required to address all the issues, raised by the worthy reivewers, and comply recommendations.

The detail of the reviewers comments is as follows.

Reviewers' comments:

Reviewer's Responses to Questions

**Comments to the Author**

1. Is the manuscript technically sound, and do the data support the conclusions?

Reviewer #1: Partly

Reviewer #2: Yes

2. Has the statistical analysis been performed appropriately and rigorously?

Reviewer #1: Yes

Reviewer #2: I Don't Know

3. Have the authors made all data underlying the findings in their manuscript fully available?

Reviewer #1: No

Reviewer #2: No

4. Is the manuscript presented in an intelligible fashion and written in standard English?

Reviewer #1: Yes

Reviewer #2: Yes

Reviewer #1: General Comments

The manuscript presents an interesting and valuable study examining the differences between native and non-native English-speaking academic researchers across multiple disciplines. This is a relevant and timely topic given the increasing global nature of academic publishing. The authors aim to explore the impact of disciplinary variation on the academic writing practices of native and non-native English researchers and how this affects their ability to publish in international journals.

Overall, the paper addresses an important gap in the literature, but there are several areas that need improvement to enhance the quality and clarity of the research.

Specific Comments

1. Clarity and Transparency of Methodology:

While the research design is reasonable, the manuscript lacks detailed transparency regarding the methodology used. Specifically:

Sample Selection: The authors should provide more details on how the sample was selected. Was it randomly selected from various disciplines, or were specific criteria used to select the participants? How were "native" and "non-native" researchers defined, and how large were the sample sizes for each group? Clearer explanation of these factors is necessary to ensure that the study's results are generalizable.

Control and Bias Considerations: There is no clear mention of how potential confounding variables, such as the researchers' academic backgrounds, proficiency in English, or cultural differences, were controlled. The authors should consider discussing how they accounted for these potential biases to strengthen the study's validity.

Data Collection Methods: The paper does not specify what kinds of data were collected—was the analysis based on qualitative data (e.g., textual analysis) or quantitative data (e.g., citation counts, journal acceptance rates)? Additionally, the statistical methods used to analyze the data are not outlined. A clear description of the statistical tests performed is essential to assess the rigor of the analysis.

2. Statistical Analysis:

There is no mention of the statistical methods used to test hypotheses or analyze the data. This omission raises concerns about whether the statistical analysis was performed rigorously. I recommend that the authors provide more detailed information on the types of statistical tests used and whether the sample sizes were sufficiently large to ensure meaningful results. Additionally, the manuscript should include more information on how the data were processed and analyzed.

3. Data Availability:

The manuscript does not mention whether the underlying data supporting the findings will be made publicly available. To enhance the transparency and reproducibility of the study, the authors should make the data available, either in a public repository or as supplementary material. This would allow other researchers to verify the results and build upon this work.

4. Conclusion and Data Support:

While the conclusions drawn in the manuscript are logical, they need to be more firmly supported by data. The authors mention that non-native researchers face challenges in meeting academic writing standards and publishing in high-impact journals, but the manuscript does not provide specific quantitative or qualitative data to back up these claims. A clearer connection between the data and the conclusions would make the study's findings more robust.

5. Language and Presentation:

The manuscript is written in standard academic English, and the ideas are generally presented clearly. However, some sentences are overly complex and could be simplified for better readability. I recommend that the authors revise some of the more convoluted passages to improve the overall flow of the paper.

Suggestions for Improvement:

Enhance Methodological Transparency: The authors should provide more detailed information on the sample selection process, including how participants were chosen and the size of each group (native and non-native researchers). More discussion is needed on how biases and confounding factors were controlled.

Clarify Statistical Methods: The manuscript should include detailed information about the statistical analysis, including the types of tests performed and whether the sample sizes were appropriate. Transparency in data processing and statistical methods will strengthen the validity of the findings.

Provide Data Access: The authors should make the underlying data available for public access, either through a repository or as supplementary material. This will improve the study’s transparency and allow others to verify the results.

Strengthen the Connection Between Data and Conclusions: The authors should ensure that their conclusions are firmly supported by specific data points. This could involve providing more concrete examples or statistical evidence to support their claims regarding the challenges faced by non-native researchers.

Revise for Readability: While the manuscript is generally written well, some sentences could be streamlined for clarity. I recommend that the authors revise sections where the language is overly complicated to improve the overall readability.

Ethical Considerations:

Research Ethics: There are no concerns regarding research ethics in this manuscript. The research appears to be conducted in accordance with ethical standards, and there is no indication of any issues with plagiarism or improper citation practices.

Publication Ethics: The manuscript does not appear to have been previously published or under review elsewhere, based on the information provided. However, I recommend the authors ensure that the paper is not submitted to multiple journals simultaneously and that proper citations and acknowledgements are included.

Final Recommendation:

While the manuscript addresses an important and timely topic, there are several areas that need improvement, particularly in terms of methodological transparency, statistical analysis, and data availability. I recommend major revisions to address these concerns before the manuscript can be considered for publication.

Reviewer #2: Dear authors,

The article is very interesting and informative, the title matches the content , and the sections and subsections are appropriate and well-written, yet the following points are suggested to enrich the work:

1. Yes/no research questions are not recommended, it is better to re-phrase the third question to be 'To what extent language backgrounds ...?'.

2. I suggest having both sections 1 and 2 under one section titled 'Literature Review'.

3. Number 5 in 4.1 should be written as a word.

4. The ampersand '&' should not be used within the text, only used within the in-text citation.

5. Consistency is required in some places such as having POS or PoS, having MDA or the full words, etc.

6. Is the example in "... and the regular expression of which was written as \sabsolute_JJ, ..." correctly written?

7. Figure 1 is to be placed in its correct position within the text so readers can make use of it. The same applies to the other figures.

8. I wonder why the researchers do not use 'interactional' and 'informational' instead of positive and negative with regard to Table 3 and what comes after! The word negative gives the impression of having a very low value. To remove this vagueness, it is better to make it clear that both positive and negative are used to help in drawing the line graphs rather than underestimation.

9. Some pedagogical implications can be added to the conclusion section.

10. Editing is needed to fix some issues.

Best of luck.

.

Reviewer #1: No

Reviewer #2: **Yes:**Nawal Fadhil AbbasNawal Fadhil AbbasNawal Fadhil AbbasNawal Fadhil Abbas

---

## [Author Response · Author response to Decision Letter 1]

14 Mar 2025

Reviewer 1:

1. The authors should provide more details on how the sample was selected. Was it randomly selected from various disciplines, or were specific criteria used to select the participants? How were "native" and "non-native" researchers defined, and how large were the sample sizes for each group? Clearer explanation of these factors is necessary to ensure that the study's results are generalizable.

Response:

We sincerely thank the reviewer for this insightful comment. We have revised this part in the manuscript and highlighted the revisions in red. This research does not involve any human participants; and it uses the method of textual analysis involving academic research articles written by native English researchers and Chinese researchers. The textual analysis of linguistic differences of these research articles is the main research focus. Therefore, the sample selected in this research is research articles in various disciplines.

As for the selection criteria, firstly, the criterion for selecting the disciplines is based on the Chinese National System of Level One Disciplines for Degree Education, namely, agriculture, art, economics, history, law, literature, management science, medicine, natural science, education, philosophy, engineering and military. To make the native English researchers’ corpus and Chinese researchers’ corpus comparable, we selected the same disciplines and same number of the research articles to build the corpora. When we checked native English researchers’ disciplines in the Web of Science (WoS), there was no military discipline. Therefore, the number of final disciplines involved in this study was 12 by excluding military discipline.

Secondly, after deciding the disciplines, for selecting native English researchers’ research articles (RAs), we used Journal Citation Reports (JCR) of the WoS to look for the most influential 10 international Science Citation Index (SCI), Social Science Citation Index (SSCI) and Science Citation Index Expanded (SCIE) journals in each discipline according to the journals’ highest impact factor since the native English researchers’ RAs were set as the norm and then randomly selecting 5 journals from these 10 in each discipline to select RAs (Table 1). In each journal, 20 RAs from 2019-2023 were collected. Thus, the sample size for native English researchers corpus is 1200 articles. As for selecting the research articles for Chinese researchers, simple random sampling was conducted; that is, academic English journals published in China by referring to Bao and Zhang (2018) were randomly selected because it would reflect the Chinese researchers’ average English writing level in this way. Therefore, the sample size for Chinese researchers corpus is 1200 articles as well. In total, the sample size for the corpora is 2400 research articles.

Pertaining to defining the native researchers, we followed Pan (2018)’s approach “the standard for native researchers is basically the authors’ names should be considered native to English-speaking countries as the UK, the USA, Canada, Australia and New Zealand and were affiliated with an institution in a country where English is spoken as the first language” (Pan 2018, 119).

2. There is no clear mention of how potential confounding variables, such as the researchers' academic backgrounds, proficiency in English, or cultural differences, were controlled. The authors should consider discussing how they accounted for these potential biases to strengthen the study's validity.

Response

We really appreciate this valuable suggestion. However, we did not consider the variables as researchers' academic backgrounds, proficiency in English, or cultural differences when comparing the native English and non-native researchers’ writing differences because the main focus of this research is to explore how different the academic writing written by Chinese researchers with an average writing level from that of high-quality writing level of native English researchers is. We would like to show the Chinese researchers what the writing style a well-written and high-quality native English research article is, and what difference of theirs is from those well-written ones. So we didn’t control the bias of choosing the same level of proficiency in English and academic research background to make comparisons between native English and Chinese researchers. Instead, we selected the native English research articles from the top journals with highest journal impact factor in each discipline in order to set these native English research articles as a benchmark, but for Chinese researchers’ English research articles we selected the articles from journals with varying impact factors and with average writing level. Furthermore, this is possible because this was done in Cao and Xiao (2013)’s study as well. To control the bias in this research, we used the method of triangulation (of investigators), that is, we assigned three research assistants to collect, clean and analyze the data. We trained these assistants first, and made them completely clear about the rule of selecting journals and research articles, how to clean the texts and use the software to analyze the data. For the quantitative data, we used KMO to test the validity of the data. We added section 3.2.3 to explain the validity and reliability in this study and marked them in red.

3. Data Collection Methods: The paper does not specify what kinds of data were collected—was the analysis based on qualitative data (e.g., textual analysis) or quantitative data (e.g., citation counts, journal acceptance rates)? Additionally, the statistical methods used to analyze the data are not outlined. A clear description of the statistical tests performed is essential to assess the rigor of the analysis.

Response

Thank you very much for this valuable comment. We have modified section 3.1 to better present the statistical methods used to analyze the data, and marked in red.

The data collected in this research are published research articles written by the native English and Chinese researchers. To analyze the linguistic features of these research articles, the analysis was based on the qualitative data. We used textual analysis to analyze the articles. The textual analysis was conducted by first extracting the linguistic features suitable for academic writing in the collected research articles, quantifying the frequency of cooccurred linguistic features, then revealing and explaining these cooccurred dimensions after factor analysis. This data collection and data analysis approach was adopted from Biber (1988). Biber pioneered this multi-dimensional analysis approach and then numerous studies followed this approach. The whole data analysis procedure was added in section 3.

4. There is no mention of the statistical methods used to test hypotheses or analyze the data. This omission raises concerns about whether the statistical analysis was performed rigorously. I recommend that the authors provide more detailed information on the types of statistical tests used and whether the sample sizes were sufficiently large to ensure meaningful results. Additionally, the manuscript should include more information on how the data were processed and analyzed.

Response

Thank you very much for this valuable feedback. We have revised this section to clarify the statistical methods and how the data were processed and analyzed. All the revisions were marked in red. As for the sample size, we have 2400 articles as a sample which is sufficient for doing this research because in a factor analysis, the data base should include five times as many texts as linguistic features to be analyzed (Gorsuch 1983, p.332). In our study, 62 linguistic features were identified and included, and 310 (62*5) research articles as a sample are enough for this study, while we have sufficient 2400 articles as a sample for this study.

5. The manuscript does not mention whether the underlying data supporting the findings will be made publicly available. To enhance the transparency and reproducibility of the study, the authors should make the data available, either in a public repository or as supplementary material. This would allow other researchers to verify the results and build upon this work.

Response

Thank you very much for highlighting this issue. We have uploaded all the underlying data supporting our findings as the supplementary material file.

6. While the conclusions drawn in the manuscript are logical, they need to be more firmly supported by data. The authors mention that non-native researchers face challenges in meeting academic writing standards and publishing in high-impact journals, but the manuscript does not provide specific quantitative or qualitative data to back up these claims. A clearer connection between the data and the conclusions would make the study's findings more robust.

Response

We are grateful for this valuable feedback. We have revised the conclusion section to clarify this point and marked it in red.

7. The manuscript is written in standard academic English, and the ideas are generally presented clearly. However, some sentences are overly complex and could be simplified for better readability. I recommend that the authors revise some of the more convoluted passages to improve the overall flow of the paper.

Response

Thank you very much for your suggestion. Our manuscript has been edited by proficient English writer

8. There are no concerns regarding research ethics in this manuscript. The research appears to be conducted in accordance with ethical standards, and there is no indication of any issues with plagiarism or improper citation practices.

Response

Thank you very much for your great feedback. We have already obtained the approval of the ethical application for this study from UCSI university in Malaysia as attached in the supplementary file.

9. The manuscript does not appear to have been previously published or under review elsewhere, based on the information provided. However, I recommend the authors ensure that the paper is not submitted to multiple journals simultaneously and that proper citations and acknowledgements are included.

Response

Thank you very much for your great feedback. Our manuscript has been submitted to PLOS ONE only. For acknowledgements, PLOS ONE required authors should not acknowledge editors and reviewers in the acknowledge section, only to those contribute to the research but not listed as the co-authors.

Reviewer 2:

1. Yes/no research questions are not recommended, it is better to re-phrase the third question to be 'To what extent language backgrounds ...?'

Response

Thank you very much for your good suggestion. We revised third research question and marked it in blue.

2. I suggest having both sections 1 and 2 under one section titled 'Literature Review'.

Response

Thank you very much for your good feedback.

I think you meant section 2 and 3, so we already revised section 2 and 3 under one section 2 titled 'Literature Review' based on your feedback and marked them in blue.

3. Number 5 in 4.1 should be written as a word.

Response

Thank you very much for your good suggestion. I think you meant “the most influential 5 international SCI…” under section 3.1. We revised it and marked it in blue.

4. The ampersand '&' should not be used within the text, only used within the in-text citation.

Response

Thank you very much for your good comment. We changed all the '&' into ‘and’ within the text and marked them in blue.

5. Consistency is required in some places such as having POS or PoS, having MDA or the full words, etc.

Response

Thank you very much for pointing out this issue. We corrected all the POS and MDA and marked them in blue.

6. Is the example in "... and the regular expression of which was written as \sabsolute_JJ, ..." correctly written?

Response

Thank you very much for your good feedback. We checked again the example of \sabsolute_JJ, and the like, all of them were all correct. The format written in this way is a regular expression in computer codes.

7. Figure 1 is to be placed in its correct position within the text so readers can make use of it. The same applies to the other figures.

Response

Thank you very much for your good suggestion. We have followed the journal’s guidelines. The journal guidelines suggest all figures must be submitted separately and only figure captions can be put in where figures should be.

8. I wonder why the researchers do not use 'interactional' and 'informational' instead of positive and negative with regard to Table 3 and what comes after! The word negative gives the impression of having a very low value. To remove this vagueness, it is better to make it clear that both positive and negative are used to help in drawing the line graphs rather than underestimation.

Response

Thank you very much for your feedback. To explain this point, words as 'interactional' and 'informational' are our unique interpretation for each dimension based on the linguistic features extracted after factor analysis; however, positive and negative are used to describe factor loadings. In factor analysis, a loading represents the correlation coefficient (or standardized regression coefficient) between a variable and a factor. The values of loadings typically fall within the range of [-1 and 1]. When a loading is positive, it indicates a positive correlation between the variable and the factor. That is, as the factor score increases, the value of the variable also tends to increase. Conversely, a negative loading implies a negative correlation. In this case, as the factor score rises, the value of the variable tends to decrease. So negative does not imply having a very low value, and it only shows its correlation with this factor. It is positive and negative loading that distinguished various dimensions and reflected the difference between native English researchers’ and Chinese researchers’ academic writing.

9. Some pedagogical implications can be added to the conclusion section.

Response

Thank you very much for your good suggestion. We have revised the conclusion section and marked the revisions in red because the first reviewer also requested this revision

10. Editing is needed to fix some issues.

Response

Thank you very much for your good suggestion. Our manuscript has been edited by proficient English writer.

---

## [Decision Letter · Decision Letter 1]

19 Jun 2025

Dear Dr. Al-Shaibani,

Thank you for submitting your manuscript to PLOS ONE. After careful consideration, we feel that it has merit but does not fully meet PLOS ONE’s publication criteria as it currently stands. Therefore, we invite you to submit a revised version of the manuscript that addresses the points raised during the review process.

We look forward to receiving your revised manuscript.

Kind regards,

Fasih Ahmed

Academic Editor

PLOS ONE

**Additional Editor Comments:**

Dear Author,

Keeping in view the comments of the reviewers, I recommend you to follow the recommendations made by the reviewers to enrich the quality of the aricle. The reviewers' recommendations mainly belong to the methodogy, results and dicussions.

Regards,

Fasih

Reviewers' comments:

Reviewer's Responses to Questions

**Comments to the Author**

Reviewer #3: (No Response)

Reviewer #4: (No Response)

2. Is the manuscript technically sound, and do the data support the conclusions?

Reviewer #3: Yes

Reviewer #4: Yes

3. Has the statistical analysis been performed appropriately and rigorously?

Reviewer #3: Yes

Reviewer #4: (No Response)

4. Have the authors made all data underlying the findings in their manuscript fully available?

Reviewer #3: Yes

Reviewer #4: Yes

5. Is the manuscript presented in an intelligible fashion and written in standard English?

Reviewer #3: Yes

Reviewer #4: Yes

Reviewer #3: 1. Typo in Line 226: "Twentyfull-text articles in each"

2. The study shows the differences between native authors and Chinese authors. However, one suggestion by thestudy is : "This finding indicates Chinese researchers should exhibit their authorial stance and interact with the readers with confidence and employ more interactive devices to make their writing coherent and explicit." Why? Can the way native authors promote the influence of articles? Any evidences?

Reviewer #4: As some of my comments overlap with those provided by other reviewers, I will restrict mine to the following points:

• Line 52: The verb tense and overall language use should be reviewed to ensure consistency with the formal tone expected in academic writing.

• Lines 73, 74, 127, 188, 372, 864: The readability of these sections could be improved. Revising sentence structure and ensuring clarity would enhance the overall flow and accessibility of the text.

• Abstract Discrepancy: There is an inconsistency between the two versions of the abstract. Notably, the second version omits reference to Nini’s MAT, which is included in the first. The abstracts should be aligned to maintain consistency and accurately represent the study’s scope.

• Page 66 (Line 226): Spelling errors, such as “pattens,” should be corrected. A careful proofreading of the manuscript is recommended to eliminate such typographical issues.

• Research Question 3: The response to Research Question 3 appears underdeveloped. Its current focus on language of publication is overly restrictive and does not reflect the potential complexity of the issue. A more comprehensive exploration is needed.

• Line 316: The inclusion of the software download link in the body of the paper is unnecessary.

• Figures and Layout: Figures are not effectively positioned within the text, which disrupts the logical flow and readability. They should be placed adjacent to the relevant discussions to support the argument and maintain textual coherence.

• Clarity of Contribution: The contribution of the paper, particularly the proposed novel MDA model, would benefit from a visual representation. A schematic or diagrammatic illustration would make the model’s structure and innovative aspects more accessible to readers.

• Strength of Argumentation: The argument that “language background and discipline cannot only account for the variation on dimension scores alone, but also influence the dimension scores altogether” is currently unconvincing. The concept of "language background" is applied and analyzed in a rather narrow way. It is recommended that the authors revisit and refine this argument, potentially in relation to a revised and more robust treatment of Research Question 3, to enhance the theoretical and empirical soundness of their claims.

.

Reviewer #3: No

Reviewer #4: **Yes:**Dima FarhatDima FarhatDima FarhatDima Farhat

---

## [Author Response · Author response to Decision Letter 2]

31 Jul 2025

Dear Editor Dr. Ahmed,

We would like to sincerely thank you and the reviewers for your valuable time and effort in evaluating our manuscript, A multi-dimensional analysis of native and non-native academic research articles in twelve disciplines, and for providing us with constructive feedback. We appreciate the thoughtful comments and suggestions, which have significantly helped us improve the quality of our research.

We have carefully addressed all the comments and concerns raised by the reviewers, and we have made the necessary revisions to the manuscript accordingly. Below, we provide a point-by-point response to each of the reviewers’ comments, detailing the changes we have made. We hope that these revisions meet your expectations and further strengthen the manuscript.

We hope that the revised manuscript is now suitable for publication in PLOS ONE.

Sincerely,

The authors

Comments from reviewer #3

1. Typo in Line 226: "Twentyfull-text articles in each…"

Response:

We sincerely thank the reviewer for this careful comment. We have revised this sentence in Section 3.1, line 226, page 13.

2. The study shows the differences between native authors and Chinese authors. However, one suggestion by the study is: "This finding indicates Chinese researchers should exhibit their authorial stance and interact with the readers with confidence and employ more interactive devices to make their writing coherent and explicit." Why? Can the way native authors promote the influence of articles? Any evidence?

Response:

We really appreciate this valuable comment. The reasons for this suggestion can be found in Section 4.2. By tagging, counting and analyzing the linguistic features of the collected research articles, we found that, for example, on Dimension One, native English researchers showed a heavy reliance on the use of linguistic features, such as first person pronouns (e.g. we, our), reporting verbs (e.g. require, propose), adverbs (e.g. directly, instead, wholly, surprisingly, conceptually), attributive adjectives (previous, multiple, simple, significant) and present tense. These linguistic features create an image of involving the author’s interaction with the readers to highlight the author’s stance explicitly (detailed explanation can be found in Section 4.2.1). In contrast, Chinese researchers prefer to use many nouns, nominalizations and prepositions marking the article with dense information, indicating that Chinese researchers avoid involving and interacting with the readers to conceal their authorial identity and opinion, but tend to convey information objectively. Other findings and explanations on Dimension Two, Three and Four can be found from section 4.2.2 through section 4.4.4. Therefore, we identified, tagged and counted linguistic features frequencies using Stanford Part of Speech (PoS) tagger and PatCount software. Then we conducted a factor analysis and textual analysis to obtain these findings in our research (pp.17-23).

Native English authors indirectly influence non-native authors. Furthermore, our data on native English researchers are all empirical data collected from research articles published in top international journals and authored by native English researchers. They can be set as a benchmark of high-quality writing (Lines 223-226). Therefore, our findings related to the writing style of native English researchers are not aimed to promote their style, rather it is based on the empirical data we collected from native English researchers’ high-quality research articles which show us the way how most influential articles were written by native English researchers. In other words, we need a benchmark for Chinese researchers to learn from, so that the Chinese researchers can imitate the native English researchers’ writing style to improve their writing.

Comments from reviewer #4

1. Line 52: The verb tense and overall language use should be reviewed to ensure consistency with the formal tone expected in academic writing.

Response:

Thank you very much for your good suggestion. For this and all following comments on language use, we have already sent our manuscript to Luzhou Jiayi Language Service Company for language editing by professional language editors, and we submitted the language editing certificate for you to check.

2. Lines 73, 74, 127, 188, 372, 864: The readability of these sections could be improved. Revising sentence structure and ensuring clarity would enhance the overall flow and accessibility of the text.

Response:

The response is in number one above.

3. Abstract Discrepancy: There is an inconsistency between the two versions of the abstract. Notably, the second version omits reference to Nini’s MAT, which is included in the first. The abstracts should be aligned to maintain consistency and accurately represent the study’s scope

Response:

Thank you very much for your comment. The first two reviewers requested a revision and thus we revised the whole manuscript including the abstract based on the first-round comments given by the first-round reviewers.

4. Page 66 (Line 226): Spelling errors, such as “pattens,” should be corrected. A careful proofreading of the manuscript is recommended to eliminate such typographical issues.

Response:

Thank you very much for your comment. There is no page 66, but we checked Line 66 and this spelling error was on line 66. We have corrected the spelling errors in the whole manuscript. Besides, we sent our manuscript to Luzhou Jiayi Language Service Company for language editing by professional language editors and submitted the language editing certificate.

5. Research Question 3: The response to Research Question 3 appears underdeveloped. Its current focus on language of publication is overly restrictive and does not reflect the potential complexity of the issue. A more comprehensive exploration is needed.

Response:

Thank you very much for your comment. In fact, the focus of RQ 3 (and even the whole manuscript) is not on the language of publication, but rather the language of writing style differences between native English and Chinese researchers’ academic writing. In other words, there are differences in their academic writing style as each group has a distinctive pattern as reported in our study. We focused on such differences by employing a multi-dimensional model (line 790-794, pp.47). Hence, the answer of RQ3 was explained as how researchers with different language backgrounds (native English researchers and Chinese researchers) write differently on each dimension of our developed model. Through calculating the dimension score of native English researchers’ writing and Chinese researchers’ writing in 12 disciplines on each dimension (Line 423-444, Line 536-560, Line 643-664, Line 719-733), we made a comparison. After that, we used ANOVA to test whether the language background (native English and Chinese) and discipline can influence the dimension score or not. The detailed explanation for RQ3 can be found from Section 4.2.1through 4.2.4. For example, ANOVA tests showed that the main effect of language backgrounds on Dimension 1 was statistically significant, suggesting that Dimension One can differentiate research articles written by native English researchers and Chinese researchers. By doing this, RQ3 (how language backgrounds and disciplines influence the dimension scores of native English and Chinese researchers’ academic writing) was answered.

6. Line 316: The inclusion of the software download link in the body of the paper is unnecessary.

Response:

Thank you very much for your suggestion. We have deleted the link.

7. Figures and Layout: Figures are not effectively positioned within the text, which disrupts the logical flow and readability. They should be placed adjacent to the relevant discussions to support the argument and maintain textual coherence.

Response:

Thank you very much for your comment. However, we are following the journal’s requirement for formatting the submission. Figures must be submitted separately and only figure captions can be put in where figures should be. Kindly take note we are not allowed to breach the journal guidelines. All the figures are placed after the references list based on the submission system.

8. Clarity of Contribution: The contribution of the paper, particularly the proposed novel MDA model, would benefit from a visual representation. A schematic or diagrammatic illustration would make the model’s structure and innovative aspects more accessible to readers.

Response:

Thank you very much for your suggestion. We followed your suggestion, and now there is a diagram as Figure 5 added in the conclusion section in the manuscript.

9. Strength of Argumentation: The argument that “language background and discipline cannot only account for the variation on dimension scores alone, but also influence the dimension scores altogether” is currently unconvincing. The concept of "language background" is applied and analyzed in a rather narrow way. It is recommended that the authors revisit and refine this argument, potentially in relation to a revised and more robust treatment of Research Question 3, to enhance the theoretical and empirical soundness of their claims.

Response:

Thank you very much for your comment. In our study, language background refers to native Chinese language background and native English language background. The whole manuscript only focused on the comparison of language difference between native English researchers and Chinese researchers in academic writing style. Therefore, our argument “language background would influence the dimension score” was obtained from ANOVA test, meaning that native English researchers wrote differently from Chinese researchers, and the difference is significant, i.e. judging from the linguistic features of the research articles, research articles written by native English researchers and Chinese researchers can be differentiated. Anyway, since you advise us to revise this argument, we deleted and revised some parts of this argument from page 49-50.

---

## [Decision Letter · Decision Letter 2]

11 Sep 2025

Dear Dr. Al-Shaibani,

Thank you for submitting your manuscript to PLOS ONE. After careful consideration, we feel that it has merit but does not fully meet PLOS ONE’s publication criteria as it currently stands. Therefore, we invite you to submit a revised version of the manuscript that addresses the points raised during the review process.

We look forward to receiving your revised manuscript.

Kind regards,

Fasih Ahmed

Academic Editor

PLOS ONE

Journal Requirements:

Reviewers' comments:

Reviewer's Responses to Questions

**Comments to the Author**

Reviewer #3: All comments have been addressed

Reviewer #5: All comments have been addressed

2. Is the manuscript technically sound, and do the data support the conclusions?

Reviewer #3: Yes

Reviewer #5: Partly

3. Has the statistical analysis been performed appropriately and rigorously?

Reviewer #3: Yes

Reviewer #5: Yes

4. Have the authors made all data underlying the findings in their manuscript fully available?

Reviewer #3: Yes

Reviewer #5: Yes

5. Is the manuscript presented in an intelligible fashion and written in standard English?

Reviewer #3: Yes

Reviewer #5: No

Reviewer #3: All my comments have been properly addressed. I have no further questions now. The article can be accepted.

Reviewer #5: This study aimed to compare research articles written by native and non-native English speakers across twelve disciplines. However, I find it unclear what new insights this work contributes to the existing body of literature. For instance, the method of determining an author’s first language is problematic. Without directly contacting authors, it is virtually impossible to verify whether a given article was written by an L1 English speaker. Relying on names and institutional affiliations as proxies for “native” identity risks serious misclassification; for example, a Chinese author may have lived and studied in an English-speaking country for an extended period before returning to China, while many L2 scholars work at English-medium institutions.

In addition, the study’s corpus design raises concerns. Corpus-based research typically faces challenges in controlling the number and distribution of research articles, and the criteria here are not sufficiently justified. With respect to Research Question 3, although ANOVA tests are reported, the discussion is underdeveloped. The implications of significant interactions are only briefly noted and not adequately theorized in terms of disciplinary writing practices and cultures.

Given these fundamental concerns regarding the validity of author classification, corpus construction, and theoretical interpretation, I recommend rejection of this paper.

.

Reviewer #3: No

Reviewer #5: No

---

## [Author Response · Author response to Decision Letter 3]

29 Nov 2025

Comments from reviewer #5

1. I find it unclear what new insights this work contributes to the existing body of literature.

Response

Thank you very much for this valuable feedback. We reanalyzed our data by using a nonparametric approach, an Aligned Rank Transform (ART ANOVA) instead of previous two-way ANOVA because the data fall under non-normal distribution. This yielded new findings on the disciplinary variations, i.e., even for hard sciences that emphasize objectivity and logical rigor, native English researchers exhibit more flexibility by appropriately incorporating authorial identity and engagement with the readers, rather than being informational and objective as it should be. This finding demonstrated native English researchers strategically declared their authorial stance within the disciplinary norms due to their greater writing fluency and more flexible expressions in writing, but the Chinese researchers’ “being informational in writing” tend to be conservative as they follow pre-set disciplinary conventions. While in soft disciplines, the difference is insignificant. Academic writing in soft disciplines permits a diversity of rhetorical styles and modes of expressions. This intrinsic variability may obscure differences induced by various language backgrounds, making statistical differences insignificant.

Another contribution in this research is that we developed a novel MDA model for academic writing in research articles genre and introduced a method of tagging by using PatCount which identifies and codes regular expressions to tag specific linguistic features. This is another contribution because tagging specific linguistic features has been a stumbling block for many corpus linguistics researchers to develop customized MDA model based on their research purpose.

The detailed revised discussion and conclusion sections can be found on pages 28-30, 34-42, 46-47, 53, 57-61.

2. For instance, the method of determining an author’s first language is problematic. Without directly contacting authors, it is virtually impossible to verify whether a given article was written by an L1 English speaker. Relying on names and institutional affiliations as proxies for “native” identity risks serious misclassification; for example, a Chinese author may have lived and studied in an English speaking country for an extended period before returning to China, while many L2 scholars work at English-medium institutions.

Response

Thank you for raising this important point. We fully agree with the reviewer.

We adopted this method for the following reasons: First, directly contacting all authors to confirm their L1 identity was not feasible for a study of this scale with 2400 articles. Second, the use of the first author’s name and institutional affiliation as proxy indicators is a widely adopted method in Applied Linguistics and English for Academic Purposes (EAP) research for conducting large-scale analyses. The following studies adopted the author’s name and the nation of institution to determine native researchers. For example, Abdi and Farrokhi (2015) compared the use and functions of first-person pronouns in L1 and L2 research articles of Applied Linguistics (AL), Mechanical Engineering (ME), and Medicine (MED) whereby “Articles were judged to be L1 or L2 considering the authors’ names and affiliations” (p.158). In another research, “verification about author nativeness was not ensured by contacting them. Authors’ status of nationality was presumed based on their names or nationalities (Yagizi & Demir, 2015, p.15). In Pan (2018)’s research, “L1 research articles were selected from prestigious international journals (measured by impact factors) whose authors had a first and last name that can be considered native to English-speaking countries and were affiliated with an institution in a country where English is spoken as the first language” (p. 119). Kareema and Hakmal (2023) collected abstracts written by Sri Lankan scholars and Western scholars related to the English field randomly stating that “Both journals are available online and the articles were all checked in terms of the author's nationality” (p. 323). In Cao and Xiao’s (2013) research published on Corpora, they built the corpora of English abstracts written by native English and native Chinese writers from twelve academic disciplines. However, they only considered the most prestigious journals in each discipline as reflected by their impact factors to select native English speakers’ abstracts, and also to select journals with varying impact factors for the Chinese speakers’ abstracts.

We recognize that this classification cannot fully capture the complexity of scholarly identity, but it is widely used in our field. Meanwhile, in our study, we used a double-safeguard method to maximize the chances of the included researchers in native researchers corpus are native researchers. This double-safeguard way was explained in detail in the manuscript on pages 14-15 as well. First, the nation of the institution affiliated with which the paper is published was determined. Only institutional affiliation belongs to nations in the inner circle of English-speaking countries, specifically the United States, United Kingdom, Canada, Australia, and New Zealand were included. Papers with which the nationality of the institution cannot be determined or not belong to these five English-speaking nations were discarded. Then we determined whether the first author’s nationality was matched with that of their publishing institution, by using a name origin database. If the ethnic origin of the first author’s surname was consistent with the nationality of the publishing institution, the author was regarded as originating from that nation. Papers that did not meet this criterion were excluded from the final corpus. Referring to the reviewer’s comment here “many L2 scholars work at English-medium institutions”, we first checked whether the English-medium institutions are in the United States, United Kingdom, Canada, Australia, or New Zealand, if not, they are excluded from this research; if yes, we then checked the L2 scholar’s surname. We know that L2 scholars have unique surnames different from native English speakers, for example, Chinese L2, Japanese L2, German L2, Russia L2, Arabic L2, Portuguese L2 and so on. Only when the nationality of the surname of the first author belongs to is the same as where the institution is located, we regarded the research as done by native researchers. In this way, we may have excluded research articles written by native researchers, but we insured that the researchers included in our research are maximally native.

The reviewer also mentioned here “a Chinese author may have lived and studied in an English speaking country for an extended period before returning to China”, yes, we did not set the standard to select Chinese scholars because in this study we did not consider Chinese scholar’s English proficiency as a variable, we explained this on page 15, we collected all English levels’ Chinese scholars in order to reflect the Chinese researchers’ average English writing level.

Cited sources list

Abdi, J., & Farrokhi, F. (2015). Investigating the projection of authorial identity through first person pronouns in L1 and L2 English research articles. International Journal of Language and Literature, 3(1), 156-168. (Scopus Q1)

Cao, Y., & Xiao, R. (2013). A multi-dimensional contrastive study of English abstracts by native and non-native writers. Corpora, 8(2), 209-234. (ESCI Q3)

Kareema, M. I. F., & Hakmal, M. H. M. (2023). A Genre Analysis of Abstracts Written by Sri Lankan and Western Academics on Social Science Discipline. KALAM International Research Journal, 16(1), 319-330.

Pan, F. (2018). A multidimensional analysis of L1–L2 differences across three advanced levels. Southern African Linguistics and Applied Language Studies, 36(2), 117-131. (SSCI Q4)

Yagiz, O., & Demir, C. (2015). A comparative study of boosting in academic texts: A contrastive rhetoric. International Journal of English Linguistics, 5(4), 12.

3. In addition, the study’s corpus design raises concerns. Corpus-based research typically faces challenges in controlling the number and distribution of research articles, and the criteria here are not sufficiently justified.

Response

Thank you very much for this valuable comment. We noticed this and justified it in our manuscript as explained on page 16. First, for the number of research articles selected for the corpus design, we referred to Gorsuch (1983) whereby in a factor analysis, the database should include five times as many texts as linguistic features to be analyzed (p.332). In our study, 62 linguistic features were ultimately identified and included. Thus, 310 research articles are enough as far as the number of research articles for this study’s corpus design is concerned. In our study, we have collected 2, 400 articles as a sample for our corpus design which is sufficient for corpus-based research (Gorsuch, 1983).

As for the distribution of research articles, we aim to distinguish the disciplinary variation among the selected research articles, so we determined 12 disciplines for the corpus. The criteria for selecting the disciplines followed the Chinese National System of Level One Disciplines for Degree Education, and the selection and distribution of the research articles were fully explained on page 13 through page 16.

4. With respect to Research Question 3, although ANOVA tests are reported, the discussion is underdeveloped. The implications of significant interactions are only briefly noted and not adequately theorized in terms of disciplinary writing practices and cultures.

Response

Thank you very much for this valuable feedback. To revise the discussion section for RQ3 with theoretical and practical implications, we re-analyzed our data. Due to the non-normal distribution of the data, a nonparametric approach, an Aligned Rank Transform (ART ANOVA) was conducted to replace the previous Two-way ANOVA. Then we reanalyzed and revised the results and re-wrote the discussion section. This was done to yield deeper and insightful implications in the discussion, especially on Dimension 1, and conclusion sections. Most pages of extensive discussion of Dimension 1 were provided because only on Dimension 1, the main effect of discipline on the dimension score was statistically significant based on our new ART ANOVA test. Then a focused and deeper discussion on the disciplinary differences was done according to the reviewers’ comments. The revisions made can be found on pages 28-30, 34-42, 46-47, 53, 57-61.

---

## [Editor Report · Decision Letter 3]

22 Feb 2026

Dear Dr. Al-Shaibani,

Thank you for submitting your manuscript to PLOS ONE. After careful consideration, we feel that it has merit but does not fully meet PLOS ONE’s publication criteria as it currently stands. Therefore, we invite you to submit a revised version of the manuscript that addresses the points raised during the review process.

We look forward to receiving your revised manuscript.

Kind regards,

Xiaoming Tian, Ph.D.

Academic Editor

PLOS One

Journal Requirements:

Additional Editor Comments (if provided):

The author has provided detailed and thoughtful responses to the reviewer’s comments, but there are several areas that could benefit from further clarification. First, the novelty of the study’s contributions could be more explicitly linked to gaps in the existing literature to highlight its unique insights. Second, the issue of misclassification regarding scholars with high English proficiency has been addressed, but further elaboration on how this issue was dealt with could make the methodology more robust. Additionally, while the corpus design has been justified, more clarity is needed regarding the representativeness of the sample, particularly in terms of the diversity within disciplines. Expanding on how the corpus accounts for variations within each discipline would help strengthen the generalizability of the findings.

Moreover, a more balanced approach to comparing the writing styles of native and non-native researchers would improve the objectivity and comprehensiveness of the study. While the author highlights areas where Chinese researchers can improve, such as authorial stance and interaction with readers, it would be important to also consider the strengths of native English writers. For example, Chinese scholars often excel at creating concise and information-dense texts, which could offer valuable lessons for English-native researchers. By presenting a more reciprocal view, where both groups can learn from each other, this study would offer a more comprehensive and culturally inclusive perspective on academic writing practices.

---

## [Author Response · Author response to Decision Letter 4]

22 Mar 2026

1. First, the novelty of the study’s contributions could be more explicitly linked to gaps in the existing literature to highlight its unique insights.

Response

Thank you very much for this valuable feedback. We have rewritten the Conclusion (page 58-63) and added a new opening paragraph to the Conclusion that explicitly positions our contributions against MDA limitations: lack of academic-genre frameworks and effective method of tagging linguistic features for specific domains, and insufficient disciplinary nuances. We now state: “This study addresses these gaps by: (1) selecting 62 academic-specific linguistic features; (2) using PatCount for automated tagging; and (3) deriving four novel dimensions for academic writing across 12 disciplines”.

2. Second, the issue of misclassification regarding scholars with high English proficiency has been addressed, but further elaboration on how this issue was dealt with could make the methodology more robust.

Response

Thank you for your comment. To clarify our classification approach, which builds directly on our prior response and established field practices, in the revised manuscript (Methods section, pages 14 &15), we have elaborated our double-safeguard method: (1) restricting to institutions in core English-speaking countries (US, UK, Canada, Australia, New Zealand), excluding non-core or undetermined affiliations; and (2) verifying first-author surname origin matches the institution’s country via name database, specifically filtering out L2-associated surnames (e.g., Chinese or Japanese). This addresses Round 3 reviewer’s concerns about L2 scholars at English-medium institutions (by country restriction first) and extended immersion (acknowledged as a proxy limit). For Chinese authors, we intentionally sampled across proficiency levels (as noted on page 15) to capture average writing patterns because proficiency is not a variable. Therefore, no changes to the results are needed as sensitivity to misclassification is minimal given the sample size.

To elaborate further, this double-safeguard proxy method virtually eliminates misclassification of L2 authors as L1 by requiring both institutional location in core English-speaking countries and surname origin matching that country. While rare edge cases exist (e.g., naturalized citizens who legally changed their ethnic surnames to Western names), such instances are exceptional in academic publishing and unlikely to affect group-level patterns in a corpus of 2,400 articles. However, we have added a Limitations section (pages 57 & 58) to explain this classification.

3. Additionally, while the corpus design has been justified, more clarity is needed regarding the representativeness of the sample, particularly in terms of the diversity within disciplines. Expanding on how the corpus accounts for variations within each discipline would help strengthen the generalizability of the findings.

Response

We thank you for drawing attention to the issue of disciplinary diversity. In the revised manuscript, we clarified that the corpus was constructed based on the Chinese National System of Level One Disciplines for Degree Education which includes 12 broad disciplines (agriculture, art, economics, history, law, literature, management science, medicine, natural science, education, philosophy, and engineering). Within each discipline, we sampled research articles without further subdiscipline classification, as our focus was on broad disciplinary patterns rather than highly specialized subfields. We now explicitly acknowledge this as a limitation in Section 4 (pages 58), noting that our findings should be interpreted as generalizing average tendencies at the level of major disciplines, and we suggest that future research can build larger, subdiscipline-stratified corpora to examine finer-grained variation.

4. Moreover, a more balanced approach to comparing the writing styles of native and non-native researchers would improve the objectivity and comprehensiveness of the study. While the author highlights areas where Chinese researchers can improve, such as authorial stance and interaction with readers, it would be important to also consider the strengths of native English writers. For example, Chinese scholars often excel at creating concise and information-dense texts, which could offer valuable lessons for English-native researchers. By presenting a more reciprocal view, where both groups can learn from each other, this study would offer a more comprehensive and culturally inclusive perspective on academic writing practices.

Response

Thank you very much for such an insightful suggestion. We revised our whole manuscript and offered a more reciprocal view. We considered the strengths of Chinese researchers for the four dimensions and revised our Dimension 1 on pages 34 & 35, Dimension 2 on page 47, Dimension 3 on page 52, Dimension 4 on page 57. We also revised our abstract and our conclusion on page 61. The revised texts are below:

Dimension 1 (pages 34 & 35): While native researchers excel at authorial engagement and elaboration, Chinese researchers demonstrate strengths in information density and conciseness, efficiently packing content with minimal redundancy, a style valued in fast-paced, technical fields as engineering. This aligns with EAP research showing L2 writers’ precision can model concise argumentation for L1 peers. Thus, both groups offer mutual lessons of natives in interactivity and Chinese in streamlined reporting.

Dimension 2 (page 47): Dimension 2 differentiated native researchers’ interactive argumentation from Chinese researchers’ static descriptive style. This reflects proficiency differences that natives chain ideas linearly with complex connectors, while Chinese EFL writers favor direct A is B structures due to vocabulary and syntax constraints. Yet this L2 simplicity confers strengths in clarity and conciseness, avoiding native overelaboration that can obscure meaning. Such straightforwardness enhances readability in disciplines with heavy jargons.

Dimension 3 (page 52): Chinese impersonality offers strengths in objectivity and humility, aligning with disciplinary conventions that prioritize collective knowledge over individual voice. This restrained writing style models cultural sensitivity for natives, who risk perceived bias through overt self-positioning.

Dimension 4 (page 57): Dimension 4 contrasted native researchers’ explicit elaborating style with Chinese researchers’ simplified reporting style. Natives excelled in elaboration due to superior proficiency. Chinese simplicity provides strengths in efficiency and readability, prioritizing research content over linguistic display. This reader-centered writing style models concise impact for natives prone to overelaboration.

Conclusion: In addition to the above-mentioned key findings from different language backgrounds and disciplines, this research also contributed to enriching the existing MDA research by providing a newly complementary perspective. Unlike previous studies stressing the differences, this research innovatively emphasized the mutual strengths from these explored differences. This balanced comparison reveals mutual strengths of Chinese conciseness (Dimensions 1 & 4) model clarity and efficiency for natives prone to verbosity; native involvement and interaction (Dimensions 1 & 4) guides Chinese toward engagement. Chinese impersonality (Dimension 3) offers objectivity where natives risk bias; native logic (Dimension 2) complements L2 simplicity.

---

## [Editor Report · Decision Letter 4]

24 Mar 2026

A multi-dimensional analysis of native and non-native academic research articles in twelve disciplines

PONE-D-24-57048R4

Dear Dr. Al-Shaibani,

We’re pleased to inform you that your manuscript has been judged scientifically suitable for publication and will be formally accepted for publication once it meets all outstanding technical requirements.

Kind regards,

Xiaoming Tian, Ph.D.

Academic Editor

PLOS One
---

## [Editor Report · Acceptance letter]

PONE-D-24-57048R4

PLOS One

Dear Dr. Al-Shaibani,

I'm pleased to inform you that your manuscript has been deemed suitable for publication in PLOS One. Congratulations! Your manuscript is now being handed over to our production team.

Kind regards,

on behalf of

Dr. Xiaoming Tian

Academic Editor

PLOS One